# Observations of cyanogen bromide (BrCN) in the global troposphere and their relation to polar surface O₃ destruction.

James M. Roberts[1*], Siyuan Wang[1,2], Patrick R. Veres[1,3], J. Andrew Neuman[1,2], Michael A. Robinson[1,2], Ilann Bourgeois[1,2,4], Jeff Peischl[1,2], Thomas B. Ryerson[1,5], Chelsea R. Thompson[1], Hannah M. Allen[6], John D. Crounse[7], Paul O. Wennberg[7,8], Samuel R. Hall[9], Kirk Ullmann[9], Simone Meinardi[10], Isobel J. Simpson[10], and Donald Blake[10]

1. NOAA Chemical Sciences Laboratory, Boulder, CO, USA
2. Cooperative Institute for Research in Environmental Sciences, CIRES, University of Colorado, and NOAA, Boulder, CO, USA
3. now at Earth Observing Laboratory, National Center for Atmospheric Research, Boulder, CO, USA
4. now at Université Savoie Mont Blanc, INRAE, CARRTEL, F-74200 Thonon-Les Bains, France.
5. now at Scientific Aviation, Boulder, CO., USA
6. Division of Chemistry and Chemical Engineering, California Institute of Technology, Pasadena, CA, USA
7. Division of Geological and Planetary Sciences, California Institute of Technology, Pasadena, CA, USA
8. Division of Engineering and Applied Science, California Institute of Technology, Pasadena, CA, USA
9. Atmospheric Chemistry Observations & Modeling Laboratory, National Center for Atmospheric Research, Boulder, CO, USA
10. Department of Chemistry, University of California Irvine, Irvine, CA, USA.

*Correspondence to:* James.M.Roberts@noaa.gov

**Abstract.** Bromine activation (the production of Br in an elevated oxidation state) promotes ozone destruction and mercury removal in the global troposphere, and commonly occurs in both springtime polar boundary layers, often accompanied by nearly complete ozone destruction. The chemistry and budget of active bromine compounds (e.g. $Br_2$, $BrCl$, $BrO$, $HOBr$) reflect the cycling of Br and affect its environmental impact. Cyanogen bromide (BrCN) has recently been measured by iodide ion high resolution time-of-flight mass spectrometry ($I^-$ CIMS), and trifluoro methoxide ion time-of-flight mass spectrometry ($CF_3O^-$ CIMS) during the NASA Atmospheric Tomography mission 2nd, 3rd and 4th deployments (NASA ATom), and could be a previously unquantified participant in active Br chemistry. BrCN mixing ratios ranged from below detection limit (1.5 pptv) up to as high as 36 pptv (10 s avg) and enhancements were almost exclusively confined to the polar boundary layers in the arctic winter and in both polar regions during spring and fall. The coincidence of BrCN with active Br chemistry (often observable BrO, BrCl and O₃ loss) and high $CHBr_3/CH_2Br_2$ ratios imply that much of the observed BrCN is from atmospheric Br chemistry rather than a biogenic source. Likely BrCN formation pathways involve the heterogeneous reactions of active Br ($Br_2$, $HOBr$) with reduced nitrogen compounds, for example hydrogen cyanide ($HCN/CN^-$), on snow, ice, or particle surfaces. Competitive reaction calculations of HOBr reactions with $Cl^-/Br^-$ and $HCN/CN^-$ in solution, as well as box model calculations with bromine chemistry, confirm the viability of this formation channel and show a distinct pH dependence, with BrCN formation favored at higher pHs. Gas phase loss processes of BrCN due to reaction with radical species are likely quite slow

and photolysis is known to be relatively slow (BrCN lifetime of ~4 months in mid-latitude summer). These features, and the lack of BrCN enhancements above the polar boundary layer, imply that surface reactions must be the major loss processes. The fate of BrCN determines whether BrCN production fuels or terminates bromine activation. BrCN reactions with other

halogens (Br⁻, HOCl, HOBr) may perpetuate the active-Br cycle, however, preliminary laboratory experiments showed that BrCN did not react with aqueous bromide ion (<0.1%) to reform $Br_2$. Liquid phase reactions of BrCN are more likely to convert Br to bromide (Br⁻) or form a C-Br bonded organic species, as these are the known condensed phase reactions of BrCN, and so would constitute a loss of atmospheric active Br. Thus, further study of the chemistry of BrCN will be important for diagnosing polar Br cycling.

## 1 Introduction

Photochemically-active halogen species in the lower troposphere affect the oxidizing capacity of the atmosphere via radical reactions that may either produce or destroy ozone (Saiz-Lopez and von Glasow, 2012;Simpson et al., 2015). In addition, active halogens play an important role in $O_3$ destruction when transported to the upper troposphere and lower stratosphere (UT/LS). Active-halogen species are defined as halogen atoms, dihalogen compounds, or species that have an

effective oxidation state above -1, (e.g., hypobromous acid, HOBr). A complete understanding of this chemistry requires an accounting of reactions that produce or destroy active halogens in a variety of environments, ranging from the polar icecaps, the ocean surface layer, and polluted continental regions.

Bromine chemistry in polar surface regions has been of particular interest since it was discovered that active-bromine compounds were responsible for ozone destruction in those regions (Barrie et al., 1988). The chemistry behind the initiation

and propagation of active-bromine chemistry has since been the subject of numerous experimental and numerical modeling studies and several review articles (Abbatt et al., 2012;Simpson et al., 2015;Simpson et al., 2007), and ozone destruction is a widespread and persistent phenomenon in the Arctic boundary layer (Jacobi et al., 2010;Halfacre et al., 2014;Ridley et al., 2003). The review articles describe how ozone destruction follows from the production of $Br_2$ or BrCl on ice, snow or particle surfaces, volatilization into the gas phase, and subsequent photolysis to two halogen atoms that react rapidly with $O_3$ to create

halogen oxide (XO, where X = Cl, Br, or I). Halogen oxides have number of pathways to form hypohalous acids (HOX) that can amplify the halogen chemistry through surface reactions with halide ions, for example:

$$HOBr + Br^- + H^+ \rightarrow Br_2 + H_2O \qquad\qquad (R1)$$

Several studies have shown that $Br_2$ tends not to form on sea ice surfaces (Pratt et al., 2013) because those surfaces are buffered at relatively high pH (Wren and Donaldson, 2012). The phenomenon represented by R1 has been termed the "bromine

explosion" when it accompanies the destruction of ozone (Wennberg, 1999;Platt and Janssen, 1995), and the participation of Br atoms in the photochemical cycle has been confirmed by direct observation of Br atoms (Wang et al., 2019). One of the keys to achieving a quantitative understanding of this Br chemistry is assessing the other heterogeneous reactions that HOBr can participate in. For example, there is evidence that HOBr reacting with dissolved organic matter (DOM), either on surface

snow/ice or on particles, could be responsible for bromoform ($CHBr_3$) enhancements observed in polar environments

(Carpenter et al., 2005;Gilman et al., 2010) and $CHBr_3$ and other organic bromine compounds observed in sea ice in the Antarctic winter (Abrahamsson et al., 2018). Reaction of HOBr with reduced nitrogen species to form cyanogen bromide (BrCN) could divert active bromine from the $O_3$ bromine explosion cycle.

    The existence of cyanogen halides has been known for some time (e.g., ClCN was discovered in 1851 (Wurtz, 1851)), but to our knowledge they have not been reported in the ambient atmosphere. Note that cyanogen fluoride (FCN) is known,

but FCN will not be formed in the environment due to the lack of chemical pathways that can form active F (e.g. HOF, $F_2$). In the environment, XCN compounds are formed from the reaction of active halogens, $X_2$, or HOX (where = Cl, Br, I) with reduced nitrogen species. For example,

    $X_2$ (HOX) + HCN → XCN + HX ($H_2O$)                     (R2)

This chemistry is well known in water and wastewater treatment processes where $Cl_2$, NaOCl or chloramines are added and

reduced nitrogen species are present (Shah and Mitch, 2012;Yang and Shang, 2004). ClCN, and to some extent BrCN, has been measured in chlorination systems, including swimming pools and spas (Daiber et al., 2016) and water treatment systems (Heller-Grossman et al., 1999), because the reaction of active-Cl compounds with bromide ions produces active Br. For example:

    $HOCl + Br^- → HOBr + Cl^-$                        (R3)

As in the case of Cl chemistry, a range of N-containing substrates may be involved in BrCN formation. The solubility of XCN compounds has recently been found to range from only slightly soluble for ClCN to moderately soluble for ICN (Roberts and Liu, 2019). Thus, cyanogen halides will easily volatilize from solution and should be considered as possible important participants in active-halogen chemistry in the troposphere.

    In addition to the above purely abiotic mechanisms, ClCN, BrCN and ICN are known to be produced in biological

systems through photosynthetic reactions involving $H_2O_2$, halides, and peroxidase enzymes present in neutrophils or other simple organisms (Vanelslander et al., 2012;Zgliczynski and Stelmaszynska, 1979;Schlorke et al., 2016). In those systems superoxide and hydrogen peroxide ($O_2^-$/$H_2O_2$) are used by enzymes to produce HOX compounds, and then XCN compounds by further reaction with N-containing substrates. In fact, there is evidence that several species of marine diatom have evolved to produce BrCN, and to some extent ICN, through photosynthetic production of $H_2O_2$ and its conversion to HOX and then

XCN compounds, by bromo- and iodo-peroxidases (Vanelslander et al., 2012). One hypothesis is that microalgae have evolved this capability because BrCN suppresses other competing marine microbiota in marine biofilm communities (Vanelslander et al., 2012). Thus, cyanogen halides and BrCN in particular could have both biotic and abiotic sources, linked through common chemical mechanisms: the reactions of HOBr with reduced-N compounds either in marine organisms and biofilms, or on ice/snow and particles in the polar boundary layer.

The production of methyl halides in snow packs, and in soil environments next to glaciers appears to be due to a combination of snow photochemical and biological processes (see for example (Macdonald et al., 2020;Swanson et al., 2007)). The literature implies that co-production of the bromomethanes, dibromomethane ($CH_2Br_2$) and $CHBr_3$ can be used to

distinguish between biotic (soils, marine) and abiotic BrCN sources. Production in soils next to glaciers showed ratios of $CHBr_3/CH_2Br_2$ on the order of 1-3 (Macdonald et al., 2020). Laboratory and field measurements show that marine $CH_2Br_2$
and $CHBr_3$ are co-products of the bromo-peroxidase process operating in marine diatoms (Mehlmann et al., 2020;Moore et al., 1996;Hughes et al., 2013), with $CHBr_3$ the primary product of peroxidase-HOBr chemistry, and $CH_2Br_2$ formed from $CHBr_3$ by bacterial action (Hughes et al., 2013). Abiotic HOBr chemistry, including water disinfection processes, produces $CHBr_3$ predominately (Maas et al., 2019;Mehlmann et al., 2020), so elevated $CHBr_3$ relative to $CH_2Br_2$ would be a marker for the atmospheric chemistry that initiates and perpetuates the bromine explosion. This aspect is borne out by the observations of organic bromine compounds in the Antarctic winter (i.e. in the absence of photosynthesis) where $CHBr_3/CH_2Br_2$ ratios were often 10 or higher (Abrahamsson et al., 2018). The source of active bromine in Abrahamsson et al., (2018) was hypothesized to be the dark reaction of ozone with bromide-containing solutions or ice, a process for which there is substantial experimental evidence (Artiglia et al., 2017;Oum et al., 1998;Sakamoto et al., 2018;Clifford and Donaldson, 2007).

The atmospheric loss processes of XCN compounds have not been studied extensively, but there are some limits that can be placed on them. Reactions of the radicals OH and Cl atoms with XCN compounds are likely quite slow ($<10^{-13}$ $cm^3$/molecule-s) because any direct abstraction reactions (e.g. OH + BrCN $\rightarrow$ HOBr + CN) are highly endothermic, and addition reactions to the -CN group are expected to be quite slow by analogy to HCN and $CH_3CN$ reactions (Manion et al., 2020). XCN photolysis has been estimated to range from no tropospheric photolysis of ClCN, to a BrCN photolytic lifetime of $\cong$135 days, to an ICN lifetime of $\cong$9 hrs at ultra violet wavelengths at summer time mid-latitudes (Fig. S1). The solubility and hydrolysis rates of XCN compounds have been measured under a variety of conditions (Roberts and Liu, 2019). That study showed that solubilities of ClCN and BrCN are relatively low (4.5 and 33 M/atm at 273K, respectively), and their hydrolysis reactions are known to be base-catalyzed (Gerritsen et al., 1993) and relatively slow at ambient pHs ($<1 \times10^{-5}$ $s^{-1}$ at 273K). The net effect of these features is that while ICN can photolyze relatively readily, ClCN and BrCN removal from the troposphere is likely governed by heterogeneous uptake and reaction with materials in those matrices, whether that be aerosol, cloud, or surfaces.

There have been almost no reported observations of XCN compounds in the gas phase in environmental systems in spite of the facile chemistry that forms XCN compounds in aquatic chemistry. ClCN has been observed by proton-transfer reaction high-resolution time of flight mass spectrometry (Mattila et al., 2020) in indoor air during experiments in which a home was cleaned with chlorine bleach solution, but to our knowledge, neither BrCN nor ICN have been observed previously in the atmosphere. Two factors may be responsible for the lack of ambient measurements of XCN compounds: (1) other chemical loss processes removing XCN species are likely much faster than hydrolysis, and (2) sensitive on-line analytical techniques for their measurement have only recently been available.

Here we report observations of BrCN in the global atmosphere during the second, third and fourth deployments of the NASA Atmospheric Tomography (ATom) experiment in January-February 2017, September-October 2017 and April-May 2018 (Thompson et al., 2022). The measurements were obtained aboard the NASA DC-8 research aircraft by trifluoromethoxide ($CF_3O^-$) chemical ionization with time-of-flight mass spectrometry detection (hereafter referred to as CIT-

CIMS) during ATom-1,-2,-3 and -4 and by iodide ion chemical ionization with high resolution time-of-flight mass spectrometry (hereafter referred to as the I-CIMS) (Veres et al., 2020) during ATom-3 and -4. The I-CIMS was not deployed for ATom-1 and -2. The project also provided an extensive suite of co-measured species used to assess the relative role of

biotic and abiotic BrCN production mechanisms. This paper summarizes the geographic distributions of BrCN, and presents several case studies of large BrCN enhancements. We then present evidence that shows a significant fraction of BrCN was produced from active-Br chemistry.

## 2  Experimental Methods


### 2.1 Experimental Design

The ATom experiment was an extensive project, with a wide-ranging set of chemical and physical measurements during four separate season deployments aboard the NASA DC-8 research aircraft. An overview of the ATom mission has been given elsewhere(Thompson et al., 2022), so only aspects pertinent to this work will be discussed here. Two instruments

detected BrCN during the ATom deployments; a time-of-flight mass spectrometer that employed trifluoromethoxide ($CF_3O^-$) ionization (CIT-CIMS) flew during all four deployments and a high-resolution time-of-flight mass spectrometer that employed iodide ion ionization (I-CIMS) flew only on the 3rd and 4th global circuits (ATom-3 and -4) in October 2017 and May 2018. The aircraft sampled the troposphere from 86°S to 83°N in more than 180 vertical profiles in each mission interspersed with level legs (the term 'level leg' refers to a flight segment during which the aircraft maintained a constant altitude (± a few

percent) for at least several minutes or more) that ranged in altitude from about 0.150 to 13.2 km, with missed approaches (MAs) over Arctic airports as low as 0.043 km. The term 'missed approach' refers to a flight period during which the aircraft descended into the controlled air space above a designated airfield, almost to the point of touching down, and climbed back out.

**2.2 I-CIMS**

The measurement of BrCN during ATom-3 and -4 was accomplished using a high-resolution time-of-flight mass spectrometer (Aerodyne Research, Billerica, Massachusetts) that employed iodide ion adduct ionization chemistry. The instrumental conditions were described by (Veres et al., 2020), and included control of pressure and water vapor concentration in the ion-molecule reaction (IMR) region. The resolving power of the mass spectrometer was approximately 5000 (m/Δm) in

the mass range of interest. This resolving power, coupled with the negative mass defect of Br yields a clear signal for BrCN, even for relatively small amounts (5.3 pptv) as shown in Fig 1. BrCN is detected as the $I^-$ adducts, $I\bullet Br^{79}CN^-$ and $I\bullet Br^{81}CN^-$, and those ions were well separated from other ions that appear at the same nominal mass. The combination of mass separation and isotope relative abundance provides conclusive identification of BrCN. The aircraft inlet for I-CIMS BrCN measurements is partially described previously (Veres et al., 2020). Additional information is that the inlet consisted of a 0.75 m length of

0.75cm ID PFA Teflon tubing thermostated at 35°C and had a flow rate of 6 slpm.

The instrument background for the BrCN signal was determined by overflowing the instrument inlet with scrubbed ambient air every 8 minutes for a 30 second period, and the detection limit of the measurements was determined by the variability in the residual signal at that mass. The scrubber system consisted of ambient air pumped from separate aircraft inlet, via a diaphragm pump, through a packed trap containing activated charcoal, Purafil, Puracarb AM, and Chlorosorb Ultra, and added to the main inlet periodically as an overflow. The scrubber material was independently checked for removal of halogens ($Br_2$, $Cl_2$, and BrCl) and other analytes of interest, and found to be more than 99% efficient at removing these dihalogens. We have every reason to expect that this scrubber also removed BrCN, however, we have examined zeroing periods that occurred in the middle of BrCN measurements, that show this to be the case. An example of such a period is shown in Figure S2, from the Oct 19, 2018 flight, in which several of the zero periods happened during elevated BrCN periods. The signals during those zeroing periods indicate that the scrubber was at least 92% efficient at removing BrCN. Also, all the zero periods exhibited non-varying signals at, or lower than, the adjacent ambient signals observed.

Calibration of the XCN compounds was accomplished using the method described by (Roberts and Liu, 2019) following the ATom-3 deployment. ClCN was produced by conversion of a calibrated mixture of HCN in air to ClCN in a reactor containing Chloramine-T (*N*-Chloro-*p*-toluenesulfonamide sodium salt). The concentration of this calibrated stream was verified by a Total Reactive Nitrogen ($N_r$) instrument (Stockwell et al., 2018) to be in the range of tens of ppbv at the CIMS inlet flow rates and the presence of ClCN confirmed by analysis with a high-resolution proton-transfer-reaction time-of-flight mass spectrometer (Hi-Res-PTR-ToF). The Hi-Res PTR-ToF instrument and its operating conditions have been described in the literature (Yuan et al., 2016;Yuan et al., 2017), and the compound was detected at ClCN•$H^+$, a process greatly aided by the Cl isotope distribution and negative mass defect. The I-CIMS had no discernible response to ClCN at I•$ClCN^-$ or potential fragment ions. A similar absence of sensitivity to ClCN was also found in a study of indoor air chemistry during chlorine bleach treatment (Mattila et al., 2020), where a PTR-MS was able to observe ClCN but the I-CIMS deployed in the same study did not detect it (J. Mattila personal communication, 2020). Standard streams of BrCN and ICN in zero air were produced using the pure compounds (Sigma-Aldrich and Acros Organics, respectively) placed in diffusion cells and calibrated by the $N_r$ instrument under constant temperature and pressure conditions. The conversion efficiency of the $N_r$ catalyst was found to be 98±10% for ClCN and assumed to be the same for BrCN and ICN based on the X-CN bond energies of these molecules (Table S1). The I-CIMS was calibrated at BrCN mixing ratios in the range of 18 - 68 ppbv and response was found to be linear over that range. The IMR temperature was held constant at 40°C during the sensitivity determinations and for a majority of the ambient measurements. The temperature dependence of the I-CIMS sensitivity to BrCN was determined in the laboratory to be nearly identical to that of formic acid (Robinson et al., 2022), and was used to correct the data from the flights on Oct. 25, 2017 and Oct. 27, 2017 during which the IMR was operated at 45°C. The I-CIMS sensitivity was also found to decrease with increased IMR water vapor concentration, in the manner shown in Figure S3, and the ion signals were normalized using the I•$H_2O^-$ ion. Response factors were then determined for the conditions corresponding to the ATom-3 and ATom-4 deployments. The I-CIMS was approximately 8× more sensitive to ICN compared to BrCN, which combined with the ClCN

result gives I-CIMS relative sensitivities in the order ICN>BrCN>>ClCN. Clear signals for ICN were observed in ambient air at the I•ICN⁻ product ion, but the large instrument backgrounds caused by the use of iodide as the reagent ion prevented quantitation of ICN (see section 3.1.2 below). Generally, detection limits of 1.5 pptv (3× standard deviation of the signal at zero BrCN) for a 10 s average with overall uncertainties of 0.4 pptv +25%, as calibrated, pertain to most measurements, with the exception of the flights during ATom-4 on May 12, May 17, and May 21, 2018, during which the detection limit was 6 pptv for 10 s averages, due to increased water vapor in the IMR and corresponding decreased sensitivity.

The instrument time response for BrCN was found to be fast ($\tau$~1.5 s) both during calibration experiments and in assessing the time variability of signal attributed to ambient BrCN. Several pieces of evidence argue against significant loss of BrCN inlets. 1) BrCN equilibrates rapidly in the short PFA inlet, consistent with our understanding of the behaviour of inorganic species on Teflon inlets as a function of Henry's Law (Liu et al., 2019). 2) BrCN hydrolysis is relatively slow and is base catalyzed (Roberts and Liu, 2019), and inlets used during ATom were acidic due to exposure to acidic aerosol (Nault et al., 2020), so loss of BrCN due to hydrolysis is not expected.

The potential formation of BrCN on the inlet surfaces of the instrument needs to be considered since it might involve the same chemistry that is making BrCN in the environment, i.e. the reaction of HOBr with reduced-N species on inlet surfaces. Several pieces of evidence indicate a lack of any BrCN production in our system. First, calibrations of the instrument with $Br_2$ and BrO standards before and after these missions did not result in any observable BrCN above detection limit. Second, the solubility, and therefore surface concentration, of HCN (a likely reaction partner) is quite low at the temperature at which the inlet was operated (40°C). In contrast, the inlet is more likely to collect non-volatile Br⁻ and convert HOBr to $Br_2$, as demonstrated by previous measurements in the Arctic troposphere (Neuman et al., 2010). As noted above, both of these heterogeneous processes are also pH dependent, and $Br_2$ will be much more favored at lower pHs that are likely present in the surface layers of the PFA inlet (Nault et al., 2020). The NOAA group has conducted preliminary lab tests aimed at examining the temperature and pH dependence of the $Br_2$ + HCN chemistry on ice/water surfaces, and confirmed the absence of BrCN production in the absence of a high pH aqueous surface on which it can take place. These experiments are described in the SI.

Several other halogen species were measured with good sensitivity by the I-CIMS as their I⁻ adducts: $Cl_2$, $ClNO_2$, BrCl, and BrO (see Table S2). Methods for the calibration of the I-CIMS for those species have been described previously (Neuman, et al., 2010;Osthoff, et al., 2008;Wild, et al., 2014) and were applied to the calibration of the ToF-CIMS used in this work. Significant losses of BrO on the actual I-ToF CIMS inlet were not apparent during calibrations that were performed before and after both ATom-3 and -4 deployments. In addition, the intercomparison of *in situ* BrO measurements by CIMS with Long Path Differential Absorption Spectroscopy (LP-DOAS) reported by (Liao et al., 2011), and test reported by Liao et al., (2012a), provide further evidence that once flow into a teflon inlet is properly initiated, an inlet of our type would not have significant BrO losses. Aside from a large shroud that was required to establish flow from a static air mass (Liao et al., 2012b), (not required for an aircraft inlet) the key feature of the Liao et al. 2011 CIMS inlet was a short section PFA Teflon tubing heated to 40°C and operated at high flow rates. Liao et al., (2011) report excellent agreement between the two methods for

periods of steady wind flow without local NO pollution. Our inlet is slightly longer than that CIMS inlet, but was made of the same material and operated at a high flow rate.

Laboratory tests were conducted during ATom calibrations to assess the amount of loss of BrO in the instrument inlet used in ATom. Those tests compared the signals observed when adding the BrO source directly to the IMR, with those observed with the addition of the 0.7m long ATom inlet operated at 6.3 SLPM. The results showed the ATom inlet transmitted 90 ±4% of BrO, leading us to conclude there was minimal loss of BrO. We further note that this inlet loss was accounted for in the applied calibration factor, determined with the ATom inlet in place. We have not observed evidence of any other losses of BrO on the ATom inlet. Previous work has shown that $Br_2$ and HOBr measurements are not reliable with the inlet used on the aircraft due to interconversion reactions noted above (Neuman et al., 2010), so those species were not quantified. However, we will present some qualitative data on these species in the Results section that support our observations of active Br chemistry.

## 2.3 CIT-CIMS.

HCN was measured by the Caltech chemical ionization mass spectrometer using $CF_3O^-$ reagent ion. HCN forms a cluster ion with $CF_3O^-$ and is detected at m/z 112 (Crounse et al., 2006). HCN was calibrated before and after each ATom deployment and a temperature and water-dependent sensitivity was used to convert the signal at m/z 112 to HCN concentrations after accounting for instrumental backgrounds determined by periodic zero measurements. The CIT-CIMS instrument was zeroed every 15 minutes in flight by closing instrument to ambient air sampling and adding ultra-high purity (UHP) $N_2$ gas. A second type of zero was also performed every 15 minutes by flowing ambient air through a filter containing $NaHCO_3$ coated nylon wool and palladium coated alumina pellets. Both types of zero generally agree well, except at very high ambient RH where HCN is not efficiently scrubbed by the $NaHCO_3$ filter. CIT-CIMS HCN calibration was performed in the lab using a compressed gas mixture of HCN in $N_2$, and a dilution system composed of three mass flow controllers, and additional $N_2$ gas source (liquid nitrogen, $LN_2$, boil-off). Fourier-transform infrared spectroscopy (FTIR) measurements were used to quantify, and monitor the stability of, the HCN mixing ratio in the compressed gas mixture using HITRAN HCN lines strengths. The laboratory HCN calibrations were proxied to flight data using periodic in-flight calibrations (every ~3 hours) of other species ($H_2O_2$, peroxy acetic acid, and ethylene glycol), after accounting for appropriate water vapor and temperature dependencies.

BrCN was also detected by CIT-CIMS as $CF_3O^-$ cluster ions at m/z 190 and 192. The inlet of the CIT CIMS has been described previously (Crounse et al., 2006;Allen et al., 2022) and consisted of a ~80 cm fluoro-polymer (Fluoropel PFC 801A, Cytonix Corp) coated glass tube under high flow (ambient pressure flow > 50 m/s) with axis perpendicular to direction of flight with an aft-facing cut, precluding cloud droplets and large aerosol particles from being sample in the inlet. The signal at m/z 192 occasionally contained contributions from other molecules – signals that do not appear at m/z 190. Thus, m/z 190 signal was used for quantifying BrCN, dividing the signal by its isotopic abundance of $^{79}Br$ (0.51) due to the calibration method employed. When the signal at m/z 190 was elevated, the enhancement at m/z 192 was as well, reflecting the approximately

equal abundance of $^{79}Br$ and $^{81}Br$ Independent BrCN temperature- and water-vapor dependent calibration factors for CIT-CIMS were determined in the Caltech laboratory using simple manometry in combination with FTIR spectroscopy. Five IR spectra with varying BrCN concentrations were collected as part of the calibration, and compared to the sum of the mass spectrometer (MS) signals at m/z 190 and 192. An integrated absorption cross-section for the band centered at 2198 cm$^{-1}$ was determined (2110–2250 cm$^{-1}$ band: $1.08 \times 10^{-18}$ cm molec$^{-1}$, log base-e at 298K). To the extent that other gases are present in the BrCN vapor sample, the integrated IR cross-section and the CIMS sensitivity represent lower limits, and in turn derived ambient concentrations represent upper limits. The CIT-CIMS was calibrated for BrCN over the range of 3 – 130 ppbv and the response was found to be linear over this range. The 1-sigma BrCN precision for CIT-CIMS 10 s data, as calibrated, is 2.5 pptv.

Ambient BrCN mixing ratios from CIT-CIMS were compared with those from I-CIMS for two ATom-4 flights with the highest BrCN: Apr. 27, 2018, and May 19, 2018. The BrCN mixing ratios from the CIT-CIMS are scattered against those from the I-CIMS in Fig. S4a for the entire two flights and in Fig. S4b from the data points for which the I-CIMS mixing ratio was > 3pptv. The data in Fig. S4b were fit using an orthogonal-distance regression (ODR) that assumes uncertainty in both variables. The slope of the correlation of CIT-CIMs versus I-CIMS is 0.52 ±0.01 and the intercept is -0.34. At this point it is not possible to privilege one calibration method and data set over the other, so they were harmonized by correcting the two data sets to the mean. Accordingly, the I-CIMS data were multiplied by 0.76 and the CIT-CIMS data were multiplied by 1.46. As a result, the correct uncertainties of the I-CIMS BrCN measurements were 0.3pptv + 50%, with detection limit of 1.1 pptv, and the 10 s precision of the CIT-CIMs measurements was 3.7 pptv. The I-CIMS data have approximately 5× better signal-to-noise than the CIT-CIMS. Consequently, the I-CIMS data are used in the analyses of the ATom-3 and -4 flights, and the CIT-CIMS data are used in the analyses of the ATom-2 Arctic flights, the only other instances in which the CIT BrCN was above the detection limit.

## 2.4 Additional Measurements

### 2.4.1 $NO_y$, $NO_x$ and $O_3$ measurements.

The oxides of nitrogen: nitic oxide (NO), nitrogen dioxide ($NO_2$) and total odd nitrogen ($NO_y$), as well as $O_3$, were measured by a 4-channel chemiluminescence system as described previously (Bourgeois et al., 2021). The inlets for the $O_3$ and $NO_y$ chemiluminescence instruments have been described previously (Bourgeois et al., 2020) (Ryerson et al., 1999). The inlet for the NO measurement consisted of a 1.5 m long 4mm ID. PFA tube thermostated at 30°C at a flow rate of 1 slpm, coupled to a fused silica tube 12mm ID, 450mm length, used to match the residence time with the $NO_2$ channel of the instrument. The use of PFA Teflon for NO and $O_3$ sampling is well established and losses of NO or $O_3$ due to chemical reactions would only be expected under extreme circumstances, (sampling of highly polluted air with extremely high VOCs or NO). The $NO_y$ inlet consisted of a solid gold tube situated in the aircraft inlet and heated to 300°C. The loss or incomplete conversion of $NO_y$ species in the $NO_y$ convertor is routinely checked through the addition of $NO_2$ and $HNO_3$ standards, and

found to be negligible. The estimated uncertainties for these measurements, and instrument precision, are $\pm(5\% + 6$ pptv) for $NO$, $\pm(7\% + 20$ pptv) for $NO_2$, $\pm(12\% + 15$ pptv) for $NO_y$, and $\pm(2\% + 15$ pptv) for $O_3$.

### 2.4.2 Whole Air Sampler.

    Bromoform ($CHBr_3$), dibromoethane ($CH_2Br_2$), methyl bromide ($CH_3Br$) and methyl iodide ($CH_3I$) were measured
during ATom-2, -3 and -4 using UC Irvine Whole Air Sampling (WAS). The WAS technique during airborne missions has been recently described by (Simpson et al., 2020). Briefly, air from outside the aircraft was drawn through a stainless-steel manifold by a dual-head metal bellows pump into individual evacuated 2-L stainless steel canisters that were sequentially filled to a pressure of 40 psig (roughly 3 atm). Air samples were collected at the discretion of samplers on board the aircraft. The average canister fill time was 53 $\pm21$ s (22–136 s) during ATom-2, 54 $\pm25$ s (18–181 s) during, ATom-3 and 52 $\pm24$ s
(16–198 s) during ATom-4, and the average sampling frequency was every 2.6 minutes for all missions. A total of 303 samples were collected for the two Arctic legs of ATom-2, and 1933 and 1853 samples were collected during ATom-3 and ATom-4, respectively, for an average of 149 and 143 whole air samples per flight. The filled canisters were returned to UC Irvine and analyzed using multi-column gas chromatography. Complete analytical details and calibrations procedures are given in (Simpson et al., 2020). The measurement precision is 5% for $CH_3I$, $CH_3Br$ and $CH_2Br_2$, and 10% for $CHBr_3$. The accuracy is
10% for $CH_3Br$ and 20% for $CH_3I$, $CH_2Br_2$, and $CHBr_3$. The detection limit is 0.005 pptv for $CH_3I$, 0.5 pptv $CH_3Br$, and 0.01 pptv for $CH_2Br_2$ and $CHBr_3$.

### 2.4.3 UV-visible Actinic flux measurements.

    Photolysis frequencies were determined from the Charged–coupled device Actinic Flux Spectroradiometers (CAFS)
(Hall et al., 2018). Up and downwelling actinic flux is collected by 2 pi steradian optical collectors above and below the aircraft fuselage. The signals are directed via fiber optics to spectrometers to provide spectrally-resolved fluxes from 290–640 nm. The photolysis frequencies are then calculated by applying the total actinic fluxes to molecular cross sections and quantum yields contained in the Tropospheric Ultraviolet and Visible (TUV) radiative transfer model (v5.3) (Madronich and Flocke, 1999). Typical molecularly dependent uncertainties range from 12–20%.

### 2.5 Box Model Description

    The model used to explore the formation of BrCN in the Arctic environment has been described previously (Wang and Pratt, 2017). It includes gas-phase, photolysis reactions, multi-phase reactions on both aerosol particles and snow, liquid-phase reactions in both deliquescent particles and liquid-like layers on snow and ice, dry deposition of trace gases. The liquid-
phase reactions of active-bromine compounds with $HCN/CN^-$ were added according to the reactions and rates specified in the Supplementary Material. The conditions and chemical measurements corresponding to the Apr. 27, 2018 missed approach over Utqiagvik, Alaska were used to initialize the model.

**2.6 Statistical Methods**

Statistical analyses were performed using the standard routines for averaging and ODR fitting provided by Igor Pro Version 8 software.

**3 Results**

The flight paths of the NASA DC-8 during ATom-2,-3 and -4 provide a unique opportunity to assess the global extent of BrCN, as those flight paths consisted of alternating high and low altitude legs around most of the global background troposphere with vertical profiles in between. The ATom-2 observations by the CIT-CIMS are limited to Arctic wintertime and will be presented below. Maps summarizing ATom-3 and -4 data from the I-CIMS instrument are shown in Fig. 2, where the flight paths are shown colored and sized by BrCN mixing ratios. BrCN mixing ratios ranged from below detection limit (1.1 pptv) up to 10 pptv for a 10 s average during ATom-3 and up to 36 pptv for a 10 s average during ATom-4. The highest BrCN mixing ratios observed during ATom-3 occurred during the low-level legs that dipped into the boundary layer at high latitudes (below 65°S and above 65°N) and the remaining observations were at or below detection limit which was consistently 1.1 pptv for the ATom-3 campaign. Likewise, the highest mixing ratios observed during ATom-4 occurred during low level legs at high latitudes, but the degraded instrument sensitivity during the May 12, 17, and 21 flights (detection limit 4.6 pptv) resulted in greater variability at low mixing ratios on the ATom-4 map (Fig. 2b). Two points of comparison that show the BrCN from wildfire (WF) is quite small-to-negligible. Biomass burning (BB) plumes observed either transported from Siberia (Oct 27, 2017) or off the coast of Africa during ATom-3 and -4 did not show BrCN above detection limit. Measurements with the same NOAA I-CIMS in WF plumes encountered on Aug 3, 2019, during FIREX-AQ (Warneke et al., 2023) showed BrCN to be at most 1 pptv, barely above detection limit (DL $\cong$ 0.3pptv during FIREX-AQ) in 1 min averaged data. These observations were in the middle of plumes in which HCN was over 40 ppbv (measured by the same CIT-CIMS).

The vertical dependence of BrCN mixing ratios is explored further by plotting the observations as a function of altitude (Fig. 3a and b) with altitude plotted on log scales to enhance the details of the lowest altitudes. BrCN was confined to the polar boundary layer and inspection of the DC-8 camera footage, where available, shows that all of those legs were over ice, not over open ocean or bare land, although not every polar boundary layer leg had measurable BrCN. The only locations where the aircraft was lower than 150m were takeoffs or landings and at coastal airports and during a missed approach (MA) over BRW, the airport at Utqiagvik, AK. The BRW MA during ATom-3 on Oct. 2, 2017, did not show elevated BrCN, or other indicators of halogen activation, as described below. The highest BrCN mixing ratios during ATom-3 reached 10 pptv and occurred over the Arctic on Oct. 25, 2017. Details of some of the individual flights highlighted in Fig. 3a are described in Section 3.1. The vertical profiles from ATom-4 (Fig. 3b) show that the highest BrCN was observed during the MA over BRW on Apr. 28, 2018, where it was clear that the aircraft entered the polar boundary layer. The other flights for which there was enhanced BrCN involved a combination of situations where the aircraft either reached the polar boundary layer or skimmed the top of the layer. Details of the MA over BRW and the other flights highlighted in Fig. 3b are described in Section 3.1.

Previous work that observed BrCN production from a marine diatom (*Nitzschia cf pellucida*) (Vanelslander et al., 2012) in laboratory cultures presents the possibility that the BrCN observed during ATom was emitted from marine biota. However, several aspects of the timing and location of the ATom observations argue against there being a significant biological source over most of the globe. BrCN was not observed in the marine boundary layer (MBL) over the areas of the ocean that had the highest biological productivity, as shown by the map in Fig. S5, which overlays the ATom flight tracks with the mean chlorophyll-α surface concentration as compiled by the NASA Earth Observatory (NASA, 2020). Unfortunately, this satellite product does not work over areas having significant ice cover (higher latitudes) so we cannot rule out a source from sea ice biological communities.

The CIT-CIMS Arctic winter flights from ATom-2 is the other set of observations that provide strong evidence that BrCN is a product of abiotic Br chemistry. The two flights, Feb. 18, 2017 from the Azores to Thule, Greenland, and Feb. 19, 2017 from Thule to Anchorage, Alaska, involved low level legs over Baffin Bay, the Arctic Ocean, and MAs over BRW and the Prudhoe Bay/Deadhorse airport (SCC). A map of the ATom-2 flights where BrCN was observed is shown in Figure S6 There was greatly reduced photosynthetic activity at the Northern latitudes during this time, compared to May 2018, yet BrCN and the ratio of $CHBr_3$ to $CH_2Br_2$ were highly correlated (Fig. 4) and $CHBr_3/CH_2Br_2$ reached levels previously observed in the Antarctic winter (>10) (Abrahamsson et al., 2018). The data for all points in both flights imply a background in $CHBr_3/CH_2Br_2$ of 2.7 ±0.3 based on the x-intercept in Fig. 4. Late winter photochemical production of $Br_2$ and $I_2$ has been noted in the snow pack at Utqiagvik, AK (Custard, et al., 2017;Pratt, et al., 2013;Raso, et al., 2017) and was probably the major source of $Br_2$ in this environment (Pratt et al., 2013). This active Br source participated in some $O_3$ destruction as indicated by some low $O_3$ points in the vertical profile shown in Figure S7, but BrCN was not closely anti-correlated with $O_3$. Also, BrCN and $CHBr_3$ appear to be produced from reaction of active Br with dissolved organic nitrogen (DON) and DOM, but the biogenic processing of $CHBr_3$ to $CH_2Br_2$ was greatly reduced compared to other seasons due to lower photosynthetic activity and lower temperatures.  Note that ATom-1 took place during the Austral winter (August, 2016), but unlike ATom-3 and -4, the aircraft did not travel south of 65.3°S, no $CHBr_3/CH_2Br_2$ ratios above 2.5 were observed during that flight, and no BrCN above the CIT-CIMS detection limit was observed during the entire ATom-1 campaign.

We examine detailed measurements from ATom-3 and -4 below for signs of $O_3$ destruction, and $CHBr_3$ and BrCN formation. These processes are in competition with one another, as they all depend HOBr/$Br_2$ chemistry:

$$HOBr + Br^-_{(aq)} \rightarrow Br_2 \qquad \text{Bromine explosion chemistry} \qquad (R4)$$

$$HOBr + DOM_{(aq)} \rightarrow CHBr_3 \qquad CHBr_3 \text{ formation/Active Br termination} \qquad (R5)$$

$$HOBr + DON_{(aq)} \rightarrow BrCN \qquad \text{BrCN formation/Active Br termination} \qquad (R6)$$

The product channels depend on the availability of the reaction substrates: Br⁻, DOM and dissolved organic nitrogen (DON), and on reaction conditions (e.g., pH). As a result, we would expect some coincidence in time and space, but not necessarily close correlations between $O_3$ loss, excess $CHBr_3$, and the appearance of BrCN.

### 3.1 Case Studies

The following analysis focuses on case studies of locations within the polar boundary layer that had BrCN enhancements, since BrCN was below detection limit throughout much of the troposphere. The data will be examined for: evidence of coincident BrCN appearance and $O_3$ destruction chemistry, co-variation of BrCN with other bromocarbons that might differentiate between biotic and abiotic origins, and indications of the processes that govern BrCN atmospheric lifetimes

and removal processes. The times noted in all the following sections are Coordinated Universal Time (UTC) to maintain consistency.

### 3.1.1 October 11 and 14, 2017, May 9 2018: Southern Ocean and Weddell Sea

The first detectable BrCN during ATom-3 was found as the aircraft flew over the southern high latitudes on October 11 and 14, 2017. There was also detectable BrCN during ATom-4 on the most southerly flight on May 9, 2018, so all these flights will be considered together. The October 11, 2017 flight involved a transit over the Southern Ocean from Christchurch, NZ to Punta Arenas, CL at approximately 65°S latitude. A low altitude leg at approximately 23:40 UTC descended to 172m and was over broken ice, i.e. ice with open water, as shown by the video frame in Fig. S8. The details of the BrCN and

associated measurements are shown in Fig. S9. This was the deepest BrCN-containing layer observed during ATom-3 and -4, extending up to about 0.8km. Along with the BrCN, there are slight amounts of BrCl and BrO (< 1 pptv) were detected, but $O_3$ was only slightly reduced (26 ppbv in the boundary layer vs 32 ppbv above), although there are not substantial amounts of data to indicate what $O_3$ levels would be expected in this environment at this time of year (McClure-Begley et al., 2017). HCN was depressed (50–70 pptv) in the polar boundary layer compared to the air immediately above 0.8 km altitude. The

halocarbons $CH_3Br$ and $CH_3I$ were not elevated in the polar boundary layer leg, but $CHBr_3$ was elevated relative to $CH_2Br_2$, although it did not have as sharp a profile as BrCN, due to the much longer atmospheric lifetime of $CHBr_3$ relative to BrCN as discussed in section 4.2 below. Another feature of note is that the $O_3$ level was quite constant (26.5 ±0.15 ppbv) in time/altitude within the intermediate polar boundary layer, which implies that if $O_3$ were destroyed due to Br chemistry in this environment it had happened sufficiently long ago that any sign of local photochemistry (e.g., in the layer closest to the surface)

had mixed within at least the middle of the polar boundary layer. See the Discussion section for a further description of the polar boundary layer.

The highest BrCN observed during southern hemisphere Spring was over the Weddell Sea during the Oct. 14, 2017 flight of ATom-3 at about 15:15 UTC and about 68°S latitude. This polar boundary layer leg was also over sea ice at 170 m. However, in contrast to the Oct. 11, 2017 polar boundary layer leg, this vertical profile shows that the polar boundary layer

did not have a constant potential temperature even down to the lowest altitude sampled (Fig. S10). As a result, the time profiles

of BrCN, and HCN, showed considerable variability at the lowest altitudes, and it is not clear from other key species, including BrCl, halocarbons and the vertical profile of $O_3$, how much Br-associated $O_3$ destruction had occurred within this polar boundary layer environment.

We observed the first firm evidence of $O_3$ destruction and the association of BrCN with active-Br chemistry later on the Oct. 14, 2017 flight during the polar boundary layer leg centered around 18:20 UTC ranging from 75–76°S latitude. During these vertical profiles, there was evidence of $O_3$ loss, and anticorrelation of $O_3$ and BrCN concentrations as shown in Fig. 5a and 5b. While it is difficult to identify a "background" $O_3$ level, i.e., a level of $O_3$ that would be present in the absence of local production or destruction or mixing down of stratospheric $O_3$, it seems clear from the entirety of the Oct. 14, 2017 flight that $O_3$ mixing ratios are in the range of 30–40 ppbv in this environment (over the Weddell Sea) at mid-tropospheric altitudes (2–6 km). Unfortunately, video from the DC-8 is not available for that flight. The TERRA/MODIS imagery from that date (NASA and Worldview, 2018) and nearby dates indicates that the Weddell Sea was largely frozen, but there was some indication from back trajectories that these air masses had passed over open leads. During the profiles shown in Fig. 5a and 3b, $O_3$ between 2–4 km was on the order of 32 ppbv (dashed lines in Fig. 5a and 5b) while $O_3$ at lower altitudes was sometimes below 20 ppbv and those levels were anti-correlated with BrCN. This type of profile is consistent with active-Br chemistry that involves $O_3$ destruction, and the production of BrCN from reactions involving active-Br compounds and reduced nitrogen substrates. In addition to this evidence, $CHBr_3$ was elevated relative to the other bromocarbons (Fig. S11) which is consistent with its abiotic production from reactions of HOBr with DOM in snow/ice and particle surfaces.

The May 9, 2018 flight during ATom-4 also flew over the Weddell Sea, this time during the Austral Fall. The BrCN mixing ratio approached 11 pptv (10 s avg), and coincided with high $CHBr_3/CH_2Br_2$ ratios (~3), but very little apparent $O_3$ loss (~1 ppbv) as shown in Figures S12 and S13. The aircraft was over the icepack, with broken leads (areas of open water where the ice pack has broken up and separated), during the two low legs where elevated BrCN was observed (14:15 UTC and 15:35 UTC) and there were substantial clouds present.

### 3.1.2 October 25, 2017, and May 19, 2018: Flights over the Arctic Ice Pack

The first episode in which BrCN was observed over the Arctic ice pack during ATom-3 was on the flight on Oct. 25, 2017 which was the northern-most leg from Bangor, Maine, USA to Anchorage, Alaska, USA. BrCN was elevated above the detection limit on four of the polar boundary layer legs, at levels ranging from 3.8 to 10 pptv (10s avg; Fig. S14). On each of those polar boundary layer legs, the aircraft did not reach a polar boundary layer with constant potential temperature. However, there was still some evidence for minor amounts of $O_3$ destruction (3–4 ppbv).

The May 19, 2018 flight was similar to the Oct. 25, 2017 flight in that it started in Bangor and ended at Anchorage and constituted the northern most leg of ATom-4. The interesting features of this flight are that each polar boundary layer leg (aside from the final approach into Anchorage) corresponded to elevated BrCN and some degree of $O_3$ destruction, as shown in Fig. 6. The photochemical environment during this flight was somewhat consistent due to the fact that the aircraft was traveling east to west during daylight hours creating a situation in which the photolysis rates did not change much with time

(see Fig. S15). As a result, the rates associated with Br radical chemistry (Br$_2$, BrCl and HOBr photolysis) were relatively consistent throughout the flight: jBr$_2 \rightarrow$ 2Br was in the range of 0.05 to 0.08 s$^{-1}$. The vertical profiles of potential temperature for each of the polar boundary layer legs (see Fig. S16) showed that the lowest layers had constant potential temperatures for all the legs except the one that was centered around 21:30 UTC (leg E). Each of the polar boundary layer legs during the May 19 flight displayed O$_3$ destruction, with the lowest O$_3$ on leg F averaging below 5 ppbv. Remarkably, these polar boundary

layer legs displayed averages of O$_3$ and BrCN that were highly anti-correlated, R$^2$=0.94 (Fig. 7). That this anti-correlation was observed over such a wide geographic expanse could mean the that O$_3$ destruction chemistry was uniform (due to the relatively uniform photochemical environment). This concept is consistent with observations and analysis presented by Halfacre et al., 2014;Ridley et al., 2003;Jacobi et al., 2010) that indicate that O$_3$ destruction is widespread and persistent in the springtime Arctic boundary layer. The anti-correlation of O$_3$ and BrCN is a necessary, but not sufficient support for the theory that BrCN

is formed by the same active-Br chemistry that destroys O$_3$ in the polar boundary layer. BrO and BrCl were also elevated during those low-level polar boundary layer legs (see Fig. S17), but did not anti-correlate with O$_3$, (R$^2$ = 0.11 and 0.06, respectively). In addition, Br$_2$ and HOBr signals were also elevated during the May 19 low levels legs (Figure S18), but not closely correlated with BrCN, nor anti-correlated with O$_3$, consistent with those of BrCl and BrO. The presence of those active-Br compounds means that O$_3$ destruction chemistry was active recently, since those compounds have lifetimes on the order of

minutes to ~1hr in this environment. In contrast, current evidence implies that BrCN has a lifetime of at least 1–2 days (see Section 4.2), so its concentration will indicate integrated Br-derived O$_3$ destruction over time.

The ICN signals during the May 19, 2018 flight exemplify the issues that were observed at this mass (I•ICN$^-$). Figure S19 shows several periods with large BrCN signals along with the normalized signal for I•ICN$^-$ (normalized to I•H$_2$O$^-$). ICN was sometimes observed when BrCN was observed, but they were not closely correlated. In addition, the instrument

background for ICN was relatively high and variable, and the signal in scrubbed air was sometimes slightly higher than in ambient air, often leading to apparent negative ICN mixing ratios. The high background was probably due to the fact that substantial amounts of CH$_3$I were used to make I$^-$ ions in the ion source and could be involved in neutral chemistry or causing surface artefacts which served to obscure ICN signals that might have been there, but were not discernible given the signal variability.

### 3.1.3 October 1, 2017 and April 27, 2018 Missed Approaches over Utqiagvik, Alaska.

The ATom-3 leg that started on Oct. 1, 2017 and flew from Palmdale, California, USA to Anchorage also included a MA over BRW at Utqiagvik shown in Fig. S20. It is instructive to examine the details of that flight as it stands in contrast to the ATom-4 MA which will be shown in detail below. This MA did not correspond to elevated BrCN concentrations; instead,

the BrCN values were slightly below zero due to uncertainties in the subtraction of the instrument background during a zero, yet there is some evidence that the polar boundary layer might have experienced a moderate amount of O$_3$ destruction (~10–12 ppbv). One of the main differences between the Oct. 1, 2017 MA and that on Apr. 27, 2018 was the presence of low-level clouds over BRW during the Oct. 1, episode as indicated by reduced jBr$_2$ and confirmed by visual evidence (Fig. S21). It could

be the that formation of BrCN is inhibited by the presence of clouds and/or BrCN was removed by uptake and reaction in clouds.

The last detailed episode to be explored took place on Apr. 27, 2018 during a MA over BRW, and involved the highest BrCN observed for the entire project. The aircraft got as low as 40 masl as it conducted the approach west to east over the runway that runs true east-west (see Fig. S22). The wind direction was from 100–120° and ranged from 4–14 m/s in the lowest 2.5km of the flight path, which means that the aircraft should not have sampled recent emissions from the town of Utqiagvik, as the town is mostly north and southwest of BRW. The bottom panel of Fig. S22 is an image from the video camera on the nose of the DC-8. It shows that the ocean adjacent to Utqiagvik was packed with sea ice at least 10km out, but it is clear that further off shore there was a mixture of open water and ice. In addition, the atmosphere was relatively clear of clouds and particles. The time series of the relevant chemical measurements during this period (Fig. 8) show that BrCN was confined to the very lowest altitude of the MA. The halocarbon data showed no elevated levels of $CH_2Br_2$, $CH_3Br$ and $CH_3I$, but significant enhancement ($\sim\times3$) in $CHBr_3$ at the lowest altitude compared to the layer just above. The vertical profiles of key chemical constituents and temperature are shown in Fig. 9 for the lowest 0.6km of the MA. There is an indication of a locally-impacted layer between 0.15 and 0.3 km directly above the polar boundary layer that had elevated $NO_y$ with an $NO/NO_y$ ratio of $\sim$0.5, but the lowest layer (< 0.15 km) had a constant potential temperature and had no evidence of local pollution. That lowest layer had BrCN up to 36 pptv (10s avg) and also elevated BrCl, and $O_3$ was as low as 32 ppbv, so reduced compared to the layer above the local pollution, which was about 45 ppbv at 0.35–0.6km.

As noted above, a general coincidence in time and place can be expected among BrCN formation, high $CHBr_3/CH_2Br_2$ ratios, and $O_3$ destruction because while they all depend on active-bromine chemistry, they are also in competition with each other. So, it is not surprising that the BrCN and $O_3$ levels from the Apr. 27, 2018 MA do not fit with the BrCN-$O_3$ correlation from May 19, 2018 that is shown in Fig. 7. However, the data set does show a general correspondence between the appearance of BrCN and the ratio $CHBr_3/CH_2Br_2$ as shown in Fig. 10a and 10b. The relationship appeared in both the Arctic and Antarctic flights but was most pronounced in the Arctic springtime. This correspondence is expected if $CHBr_3$ is formed from active bromine reacting with DOM, and BrCN from reaction with DON, as DON is a subset of DOM. We explore some of the possible relevant BrCN formation and destruction chemistry in the next section.

## 4 Discussion

In many of the polar boundary layer airmasses sampled, BrCN levels observed were often substantial relative to other gas phase organic bromine compounds ($Br_{org}$). For example, during the low altitude leg on the May 19, 2018 flight centered at 22:30, BrCN was 19 pptv while the sum of Br in $CHBr_3$, $CH_2Br_2$, $CH_3Br$, $CHBrCl_2$, and $CHBr_2Cl$ was just 16 pptv. Similarly, BrCN observed during the Apr. 27, 2018 MA, 36 pptv, was higher than the corresponding $Br_{org}$, 27 pptv.

The observations of BrCN in polar boundary layer air have some intriguing implications for the active Br chemistry that occurs in these environments. If there is multi-phase chemistry occurring that terminates active Br during classic "bromine

explosion" chemistry to form BrCN, then the active-Br system works differently at destroying $O_3$ and Hg than is currently thought. This impact also depends on the fate of BrCN, which is not well understood since gas phase loss processes are either known to be, or likely to be, very slow (Roberts and Liu, 2019). In addition, BrCN solubility in water is low, and its hydrolysis rates are slow at environmental pHs. Yet, circumstantial evidence (e.g., the near absence of BrCN above the polar boundary

layer) indicates BrCN has a relatively short atmospheric lifetime. The mechanism for heterogenous removal of BrCN is also unclear and could have implications for whether that chemistry could re-form active Br, or be a net sink for active Br. We discuss the possibilities for BrCN formation chemistry and some constraints and possible mechanisms of heterogeneous loss in this section.

**4.1 BrCN Formation**

As noted above, formation of BrCN has been demonstrated in a number of chemical systems both abiotic and from marine biota. The bulk of the evidence in ATom-2, -3 and -4 datasets is that BrCN formation occurs only in polar regions when there is active-Br chemistry occurring. Conversely, neither of the ATom-3 and -4 deployments observed instances where there was substantial $O_3$ destruction and no measurable BrCN present. The destruction of $O_3$, i.e. $O_3$ substantially below what

is in the lowest 1–2km of the atmosphere above, is one tell-tale sign of active-Br chemistry. Another sign of active-Br chemistry is the presence of elevated $CHBr_3$ concentrations relative to the closely-related biogenic compound $CH_2Br_2$. These markers could be considered evidence that BrCN was produced from active-Br chemistry. As a consequence, it is useful to consider what chemistry could be forming BrCN from active Br, i.e., what is the source of reduced nitrogen and how would the heterogeneous formation chemistry work.

As with all XCN compounds, it is difficult to devise a chemical scheme in which BrCN is produced via gas-phase chemistry. There are basically two reasons for this: Br atoms are either not very reactive or only moderately reactive with most organic compounds (Barnes et al., 1989), but reasonably reactive with $O_3$ ($k = 1 \times 10^{-12}$ cm$^3$/molec-s). As a consequence, reaction with $O_3$ will out-compete any organic reactions due to its higher abundance as long as $O_3$ has not been completely depleted, a condition not observed in the ATom data set. Conversely, even if there were a source of CN radicals, which is quite

unlikely, they react so rapidly (for example with $O_2$) that it is highly improbable that there could be a Br-compound that could compete with other CN reaction partners in rate or in mechanism (Manion et al., 2020).

The production of BrCN by heterogeneous or condensed phase (i.e., snow, ice, frost flower, aerosol particles) chemistry is much more plausible as condensed phase mechanisms are known. However, there are still a number of unknowns that make it difficult to construct a quantitative model. Rate constants and solubilities of the key reactants are not known in

what is presumably a liquid-like layer of water below 273K. The active-Br compounds that are involved are $Br_2$ and HOBr, but the identity and chemistry of the N-containing substrates are uncertain. The possibilities are HCN/CN$^-$, and reduced N moieties (amines, amides, amino acids) that are part of DON present in polar environments. These DON compounds can be resident on particles (Dall'Osto et al., 2017) or ice/snow surfaces that are known to collect organic materials from a number of sources (McNeill et al., 2012). These include deposited atmospheric aerosol, sea spray, frost flowers, and sea water that has

been wicked up into the ice floe. These sources by their nature are mostly DOM, and hence provide the substrates needed to produce $CHBr_3$ from HOBr through the haloform reaction. Thus, we would expect BrCN and $CHBr_3$ enhancements to be coincident in time and location, which is what is observed in our polar boundary layer data.

It is useful to examine whether reaction of HCN/CN¯ on snow or ice to form BrCN could compete with propagation of Br-explosion chemistry through known mechanisms and how conditions, particularly substrate pH, might affect BrCN
formation from that pathway. Simple calculations based on the competition of HOBr between reaction with HCN/CN¯ to make BrCN and reaction with Br¯ or Cl¯ to make $Br_2$ or BrCl, and a box model calculation with more extensive Br chemistry(Wang and Pratt, 2017) are presented in the SI, and only summarized here. The competitive reaction analysis showed a profound pH dependence with higher BrCN formation at higher pH (Fig. S23 and S24). This was due to the pH dependence of HCN solubility and HOBr/OBr¯ equilibrium, and also pH dependences of HCN/CN¯ reactions with HOBr/OBr¯. Higher Cl¯ and
Br¯ concentrations will serve to shift the pH at which BrCN becomes competitive to higher values, but the range still appears to be 6.5-8.5 pH units, a range that is applicable to matrices in polar environments. The box model with more extensive Br chemistry also showed a pH dependence, with higher BrCN favored at higher pH (Fig. S25), but more Br propagation at mid-range pH because of pH-independent Br chemistry in the model. We are not aware of any measurements of CN¯ in natural snow pack or sea ice.

The pH of condensed phases in the Arctic environment can range from fairly acidic (pH 1–2) for aerosols (Nault et al., 2020) to over pH 8 for frost flowers or sea spray aerosol (Kalnajs and Avallone, 2006). In addition, the pH of sea ice is somewhat buffered at relatively high pHs (Wren and Donaldson, 2012), providing a persistent matrix for BrCN formation from HCN/CN¯ and other reduced N compounds. So, the coincidence of BrCN with elevated $CHBr_3$ and $O_3$ depletion seen in some of the data presented here is consistent with abiotic active-Br chemistry. However, if the competition between production
of BrCN and the propagation of active Br is pH dependent, we would not necessarily expect the tight anti-correlation shown in Fig. 7 for the May 19, 2018 flight, unless the conditions in which this chemistry was taking place was relatively uniform over 1000s of kilometers. Moreover, the fact that the BRW MA data do not fit with the data from the May 19, 2018 flight might simply be due to differences in amount and pH of the DON substrate available for reaction.

The observation that dihalogens are produced by photochemistry in the snowpack (Pratt et al., 2013;Custard et al.,
2017;Halfacre et al., 2019) also presents the possibility that BrCN and $CHBr_3$ are produced by DON and DOM in that environment, the key reactions being with $Br_2$ or HOBr that are produced in the condensed phase. Since both BrCN and $CHBr_3$ are relatively insoluble they would be readily emitted to the atmosphere. Another intriguing possibility exists for the production of BrCN (and other active-Br compounds) in the snow/ice environments. The production of active Br in algae, HOBr in particular, is an enzymatic process that involves the reaction of $H_2O_2$ and Br¯ in the presence of a peroxidase. As such, HOBr
production is intimately tied to photosynthetic processes as has been shown in several laboratory studies (Hughes et al., 2013;Vanelslander et al., 2012;Moore et al., 1996). It is possible therefore, that dihalogen production, along with BrCN, and $CHBr_3$, could appear photochemical in nature, but in fact rely on the presence of photosynthetic algae, and in the case of BrCN and $CHBr_3$ the presence of DON and DOM substrates that would go along with algae in this environment. Again, the

correlations of BrCN with $O_3$ destruction argue for a substantial source of BrCN from abiotic active-Br chemistry at least some of the time.

## 4.2 BrCN Loss

As described in the introduction, gas phase loss processes of BrCN are either known to be slow (photolysis lifetime at UV wavelengths $\cong 4$ months at summer mid-latitudes) or quite likely to be slow (OH and Cl radical reaction rate constants $< 10^{-14}$ cm$^3$-molec$^{-1}$s$^{-1}$ by analogy to HCN and CH$_3$CN)(Manion et al., 2020). The heterogeneous loss of BrCN to surfaces, particles or cloud water has only been estimated based on aqueous solubilities and a few liquid-phase reaction rates, e.g. hydrolysis and reaction with n-octanol(Roberts and Liu, 2019) and those range from several weeks to 0.5 years. There are numerous references on solution phase reactions that BrCN undergoes, but we were not able to find quantitative information about reaction rates (see for example (Li and Gevorgyan, 2011;von Braun and Schawarz, 1902;Van Kerrebroeck et al., 2022)). At this point it is worth examining the ambient measurements for further indications of what the BrCN lifetime is in the polar boundary layer. Figure 11 shows vertical profiles of potential temperature, ozone, BrCN and the halocarbons CH$_2$Br$_2$ and CHBr$_3$ from the May 19, 2018 polar boundary layer leg at 2230 UTC that was discussed above in Section 3.1.2. There was very little BrCN above the polar boundary layer, and this was a general feature of all the instances where BrCN was observed. It is clear that BrCN shows a much sharper decrease with altitude than the bromocarbons (CHBr$_3$, CH$_2$Br$_2$) that have lifetimes on the order of 2–4 months at this latitude and season (Papanastasiou et al., 2014;Hossanini et al., 2010), thus BrCN must have a shorter lifetime than those compounds.

Most of the observations of elevated BrCN occurred within layers that had constant potential temperature and appeared relatively well-mixed, i.e. O$_3$ was depleted but constant. However, the interpretation of these measurements is complicated by the fact that our observations did not extend to the surface. Observations in polar boundary layers show that there are often very shallow stable layers (a few to tens of meters) close to the surface(Anderson and Neff, 2008) below a relatively well-mixed layer a few 100 meters to a kilometer in depth. This intermediate layer is usually capped by a strong temperature inversion and there are often clouds in the vicinity of the inversion (Serreze et al., 1992). Several model studies have shown that very low diffusivities (1 $\times 10^{-3}$ m$^2$/sec) best describe atmospheric observations in and above these layers (Zeng et al., 2006;Wang et al., 2008). By analogy to molecular diffusion, this corresponds to transport times out of the boundary layer through this inversion that are quite long (> weeks). The other factor that comes into play for many of these polar boundary layer legs is that many occurred over open leads in the ice, which are the source of vertical transport due to the effects of warmer water(Serreze et al., 1992). As a consequence, transport of BrCN out of the top of polar boundary layers is likely more limited by synoptic disturbances like frontal passage or ice lead formation. From these features we can roughly estimate that BrCN could have a lifetime as short as a day or two, but probably not longer than 10 days in the polar boundary layer, the upper limit being based roughly on the frequency of synoptic disturbances in this environment. It follows from this, and the slowness of gas phase loss processes, that heterogeneous reactions limit the BrCN atmospheric lifetime.

The rates of these possible condensed-phase reactions can be estimated in the same type of analysis employed by Roberts and Liu (2019) given a few assumptions about the polar boundary layers encountered in this work and Henry's Law solubilities. Deposition of a chemical species within a boundary layer can be thought of as two processes happening in series: physical transport to the surface, and chemical reaction at the surface (see for example (Cano-Ruiz et al., 1993)). These can be parameterized with a simple resistance model:

$$v_d = 1/(R_t + 1/(\gamma<c>/4)) \tag{Eq1}$$

where the total resistance is $=1/v_d$, with $v_d$ being the deposition velocity, which can also be expressed as the mixing layer height, $h$, divided by the lifetime of the chemical species, in this case $t_{BrCN}$, and $\gamma$ is the surface uptake coefficient and $<c>$ is the mean molecular speed.

Our study lacks the detailed polar boundary layer dynamics measurements needed to fully evaluate $R_t$, the resistance due to turbulent diffusion in the lowest altitude legs. However, there are several aspects of the observations that allow us to make a useful limiting estimate, using the May 19, 2018 22:30 polar boundary layer leg as the prime example (Fig. 11). First, there were open leads (Fig. S26) that are associated with polar boundary layer mixing and relatively high mixing heights (Serreze et al., 1992;Moore et al., 2014). Second, the fact that potential temperature is constant with altitude below 300 m implies the polar boundary layer is well mixed at least as low as the aircraft sampled. Third, there were substantial vertical wind velocities measured by the gust probe aboard the DC-8 aircraft during this polar boundary layer leg, also shown in Fig. S26. All of these features support the assumption that, at least in this instance, turbulent resistance $R_t$ is small relative to the resistance due to chemical uptake, at least in the presence of open leads. This is consistent with estimated mixing times of several hours deduced in studies of ozone and mercury in polar boundary layers impacted by open leads (Moore et al., 2014).

Consequently, if chemical uptake is the limiting loss process, we can estimate the liquid phase loss rate of BrCN ($k_l$) from Equations 1 and 2. The resistance due to chemical reaction is $1/(\gamma<c>/4)$ and the uptake coefficient ($\gamma$) can be estimated from the following equation assuming surface accommodation is efficient (Kolb et al., 1995)

$$\gamma = 4HRT(k_lD_a)^{1/2}/<c> \tag{Eq2}$$

where $H$ is the Henry's coefficient, $k_l$ is the first order liquid phase loss rate, $D_a$ is the diffusion coefficient of the reactant in solution, and $RT$ are the gas constant and temperature. Similarly to the analysis of HCN solubility, the parameters needed to estimate $k_l$ will need to be extrapolated below 273K assuming the surfaces are liquid-like layers. In the case of BrCN, we use recent solubility measurements (Roberts and Liu, 2019) and assume $H$ has the same temperature dependence (i.e., same enthalpy of solution) in the liquid-like layer below 273K. Diffusion coefficients also vary as 1/T, and although there are not measurements of $D_{BrCN}$, we estimate it is $0.5 \times10^{-5}$ cm$^2$/s at 263K by analogy to HCN and the temperature dependence of diffusion coefficients(Tyn and Calus, 1975). Thus, the first order loss rates of BrCN in solution that correspond to polar boundary layer lifetimes range from $1.2 \times10^{-2}$ s$^{-1}$ for a lifetime of 1 day, to $1.2 \times10^{-4}$ s$^{-1}$ for a lifetime of 10 days. This range is

faster than BrCN hydrolysis, which can be extrapolated to be approximately $1.5 \times 10^{-5}$ s$^{-1}$ at 263K(Roberts and Liu, 2019). Given that any reactants other than water would be present in micromolar to nanomolar concentrations, this range of first order loss rates corresponds to second order rate constants that are in the range of $10^4$–$10^7$ M$^{-1}$ s$^{-1}$.

The possibility that BrCN could react with halide ions (Cl$^-$, Br$^-$, or I$^-$) and reform a dihalogen (e.g. XBr) should be considered, as those reactions would serve to propagate Br explosion chemistry. Reactions of BrCN with I$^-$ and Br$^-$ have been studied in solution (Nolan et al., 1975) and it was found that the rate limiting step in BrCN + I$^-$, which probably forms IBr, is relatively slow, 0.106 M$^{-1}$s$^{-1}$ at 298K. The solution-phase reaction eventually formed I$_3^-$, which would not be favored at typical I$^-$ concentrations in polar environments. The reaction of BrCN with Br$^-$ was estimated to be quite a bit slower ($\sim 10^{-8}$ M$^{-1}$s$^{-1}$) based on rate measurements of Br$_2$ + HCN $\rightarrow$ BrCN + Br$^-$ + H$^+$, and the measured equilibrium constant that was also given (Nolan et al., 1975):

$$K = \frac{[BrCN][H^+][Br^-]}{[HCN][Br_2]} = 6 \times 10^8 \text{ M} \qquad \text{(Eq 3)}$$

We attempted to form Br$_2$ in a laboratory experiment at room temperature in which saturated NaBr solution was exposed to 150 ppbv of BrCN (see details in the SI), and no observable Br$_2$ resulted (upper limit of 0.1% of BrCN), consistent with the above estimate that the reaction BrCN + Br$^-$ $\rightarrow$ Br$_2$ +CN$^-$ is quite slow. Although it has not been measured to our knowledge, we can expect the reaction of BrCN with Cl$^-$ to be quite slow as well, given established trends of halogen reactions.

The reaction of BrCN with active-halogen species, particularly HOCl or HOBr;

$$\text{HOCl (or HOBr)} + \text{BrCN} \rightarrow \text{BrCl (or Br}_2\text{)} + \text{HOCN} \qquad \text{(R4)}$$

might also occur on snow/ice or particle surfaces. While there is no direct evidence for this, the addition of HOCl or Cl$_2$ to treated water where BrCN was present was observed to accelerate BrCN loss (Heller-Grossman et al., 1999). Such a reaction of HOCl or HOBr, combined with formation reactions discussed above would lead to no net gain or loss of active halogen to the polar boundary layer.

There are a number of other reactions that BrCN is known to undergo, aside from simple hydrolysis, but none of them generate active Br. BrCN is known to react with R-OH- or R-SH-containing compounds and substituted amines (Siddiqui and Siddiqui, 1980;Kumar, 2005), with the net reaction yielding cyanate, thiocyanate or nitrile functionalities and the Br ending up either as Br$^-$ or as a Br-carbon compound. To our knowledge, the rate constants of these reactions have not been measured, but are likely to be reasonably fast given that many of these reactions are used for synthesis or column preparative derivatization (see for example (March et al., 1974)). The key point of emphasis here is that none of these reactions leads to the reformation of an active-Br compound.

## 5 Conclusions

Two different instruments measured BrCN during the NASA ATom deployments, a $CF_3O^-$ ToF CIMS and an $I^-$ ToF CIMS, along with two other active-bromine compounds BrCl and BrO during the NASA ATom mission deployments 3 and 4 in 2017 and 2018. The ATom-1 and -2 measurements from the $CF_3O^-$ ToF CIMS showed that BrCN was present above detection limit ($3\sigma$ of the 10 s precision = 11 pptv) only in the polar boundary layer during the Arctic winter. This Arctic BrCN approached levels as high as 38 pptv and was correlated with high ratios of $CHBr_3$-to-$CH_2Br_2$, a marker for active-Br chemistry in the absence of photosynthesis. The elevated BrCN and $CHBr_3$ have a common origin in the condensed-phase reactions of active bromine (HOBr, $Br_2$) with dissolved organic matter (DOM) and dissolved organic nitrogen (DON). In this late winter environment, there is greatly reduced photosynthetic activity, but $Br_2$ can be formed from photochemistry in the snow pack (Custard et al., 2017), hence the relatively tight correlation of BrCN and $CHBr_3/CH_2Br_2$ arises due to the fact that DON is a subset of DOM.

The ATom-3 and -4 measurements revealed elevated mixing ratios (> 3 pptv) of BrCN in both polar boundary layers during both seasons (Spring and Fall), October 2017, late April – May, 2018. Mixing ratios of BrCN up to 36pptv (10s avg) were observed during a MA over BRW airport on April 28, 2018, at Utqiagvik, Alaska. These observations also support the conclusion that least some of the BrCN came from active-Br chemistry reactions on snow, ice and/or particle surfaces, with the -CN group resulting from either $HCN/CN^-$ or other reduced nitrogen species from oceanic or cryospheric sources. Instances where substantial BrCN was observed corresponded to other indicators of active-Br chemistry: $O_3$ depletion, elevated $CHBr_3$ relative to $CH_2Br_2$, and usually elevated BrO and BrCl mixing ratios. A simple competitive reaction model of the liquid phase reactions of $HOBr/OBr^-$ with $HCN/CN^-$ or $Cl^-/Br^-$ and a box model of Br chemistry both show that BrCN formation can compete with active-Br propagation at higher pHs (pH>6). Surface pH is a somewhat uncertain property of the liquid-like layers that are on the surfaces of aerosol, ice, and frost flowers, and it could also be that during a given $O_3$ depletion episode the chemistry could evolve to produce BrCN on basic surfaces such as frost flowers and brines, or as surface acidity of snow and ice is depleted. We also note that sea ice surfaces are buffered with respect the deposition of acids, so can remain at the higher pHs characteristic of sea water (Wren and Donaldson, 2012), which will favor BrCN formation chemistry.

The higher mixing ratios of BrCN observed at times during polar boundary layer legs make it a potentially important contributor to the active-Br cycle. Total inorganic Br values are typically on the order of a few tens of pptv in areas where bromine explosion chemistry is taking place (Simpson et al., 2015). In addition, very short-lived halocarbon species that contain bromine also usually amount to a few tens of pptv Br, as indicated by bromocarbon mixing ratios reported in this paper. The BrCN concentrations observed in this study were often in the same range as these important Br reservoirs.

BrCN was confined to the polar boundary layer and vertical profiles showed a sharp cut-off at the level of the temperature inversion strongly implying that BrCN has a short lifetime (1–10 days) in this environment. Given the relatively low aqueous solubility and slow hydrolysis reaction rates of BrCN, these vertical profiles imply BrCN must undergo more rapid condensed-phase reactions ($10^4 - 10^7$ $M^{-1}$ $s^{-1}$) that lead to relatively efficient uptake and loss in the polar boundary layer

environment. The presence of BrCN indicates a loss of active bromine from the system since known condensed phase loss mechanisms of BrCN result in either $Br^-$ or C-Br bonds. Preliminary laboratory experiments showed that BrCN does not reform active bromine when reacted with solution phase $Br^-$. Further intensive ground-based observations of BrCN with other active-Br compounds, and laboratory exploration of BrCN liquid phase reactions will lead to better characterization of the processes that control halogen-induced $O_3$ loss in polar regions.

## 6 Code availability

Statistical analyses were performed using the standard routines for averaging and ODR fitting provided by Igor Pro Version 8 software. The box model code is available on request.

## 7 Data availability

Data are available in the main text and supplementary materials and from the Atmospheric Tomography mission archives in the Oak Ridge National Laboratory Distributed Active Archive Center (ORNL DAAC) at https://doi.org/10.3334/ORNLDAAC/1925

## 8 Supplement link

## 9 Author contributions

JMR wrote the paper with assistance from co-authors. JMR, PRV, JAN, MAR, JDC, and POW worked on BrCN calibration. SW performed model analyses on Br chemistry. PRV, JAN, IB, JP, TBR, CRT, HMA, JDC, POW, SRH, KU, DB, and SM conducted the ATom measurements.

## 10 Competing interests

The Authors declare they have no competing interests

## 11 Disclaimer

Mention of tradenames and brands is for information purposes only and does not imply an endorsement by the authors.

## 12 Acknowledgements

We thank William D. Neff and Matthew Shupe for helpful discussions. We acknowledge the ATom Science Team who contributed to this mission. We also gratefully acknowledge the NASA and ESPO project personnel who participated in this campaign. This work was supported by; The NOAA Cooperative Agreement with CIRES, NA17OAR4320101.

The ATom project, an EVS-2 Investigation under NASA Research Announcement (NRA) NNH13ZDA001N-EVS2, Research Opportunities in Space and Earth Science (ROSES-2013) and funded through NASA Agreement NNH15AB12I to NOAA.

Contributions from Caltech were funded through NASA agreements NNX15AG61A and 80NSSC21K1704.

The National Center for Atmospheric Research, which is sponsored by the National Science Foundation under Cooperative Agreement 1852977.

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

**Figures**

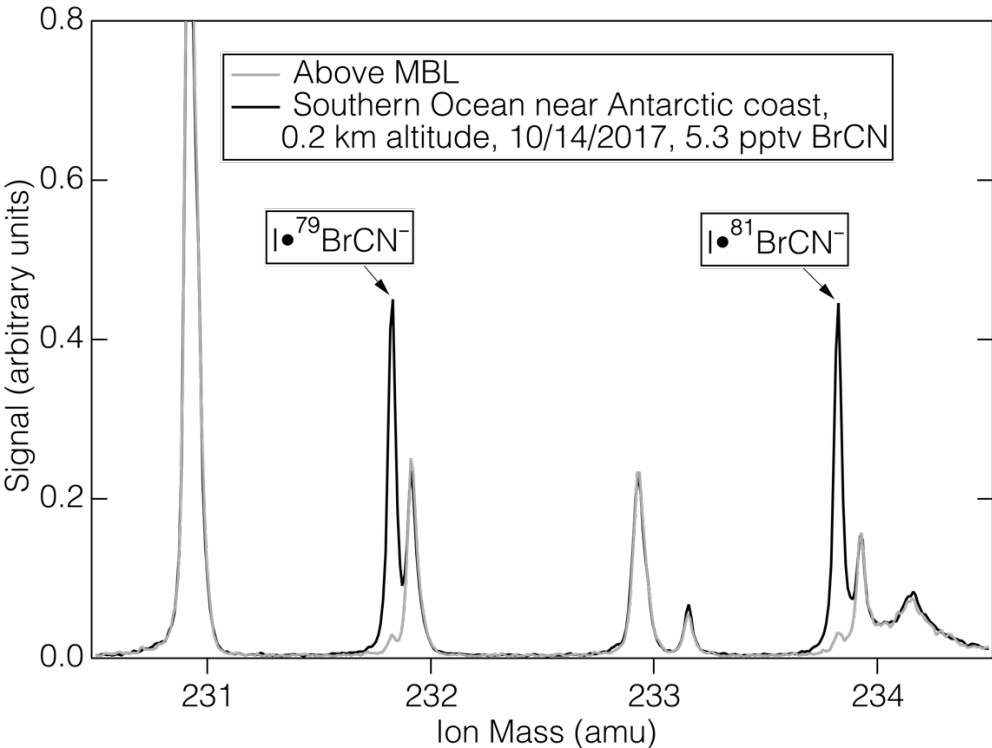

**Figure 1: The mass spectra of ambient air (1 s average) during sampling in the marine boundary layer near Antarctica during**
**ATom-3 (black) and that of ambient air above the MBL (grey).**

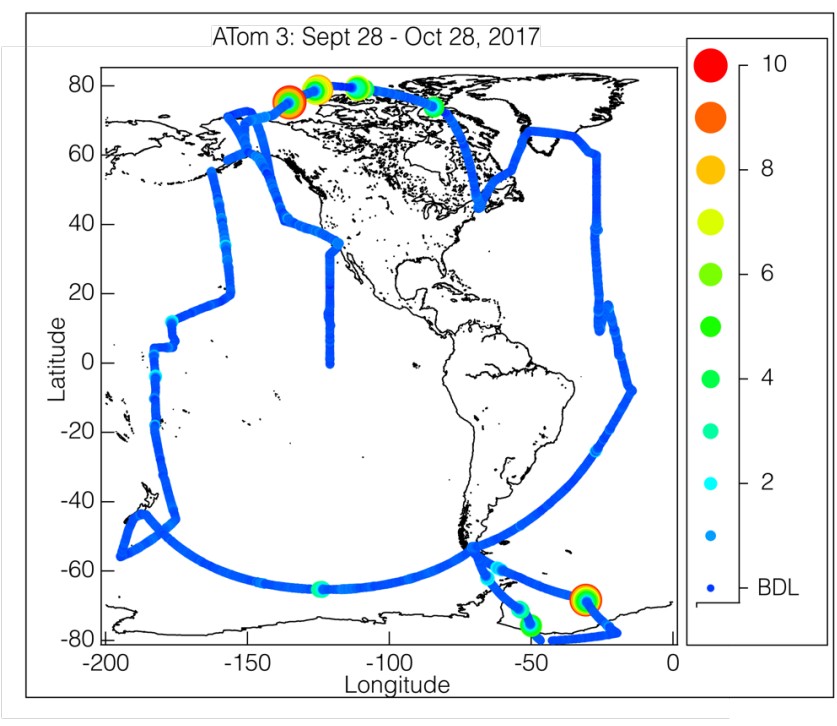

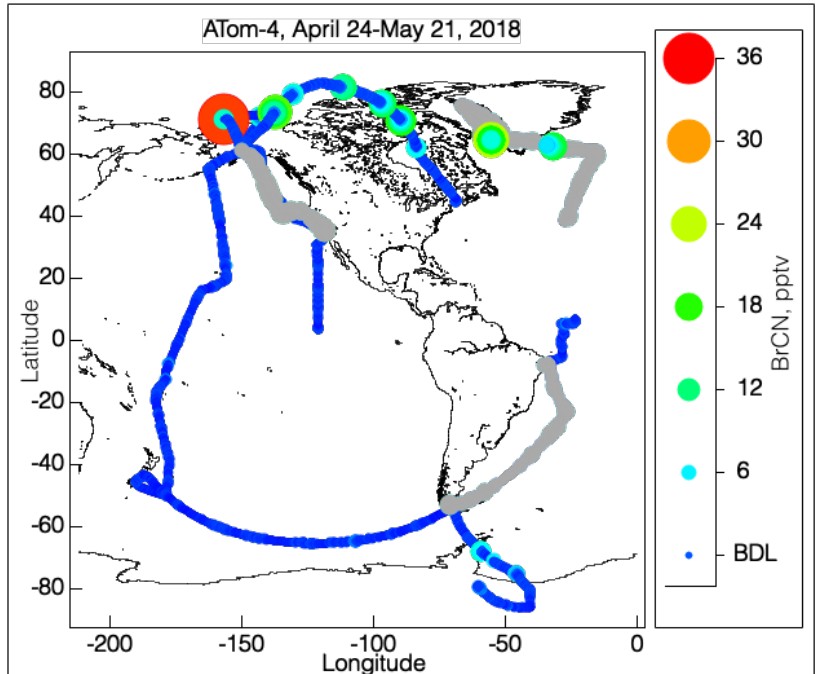

**Figure 2. Maps of the BrCN mixing ratios (10s averages) observed during ATom-3 (top panel) and ATom-4 (bottom panel) colored and sized by mixing ratio, as shown by the adjacent scales. The detection limits for most flights were 1.5 pptv, except for the observations made on May 12, 17 and 21, 2018 that had higher detection limit (6pptv) and are shown in grey.**

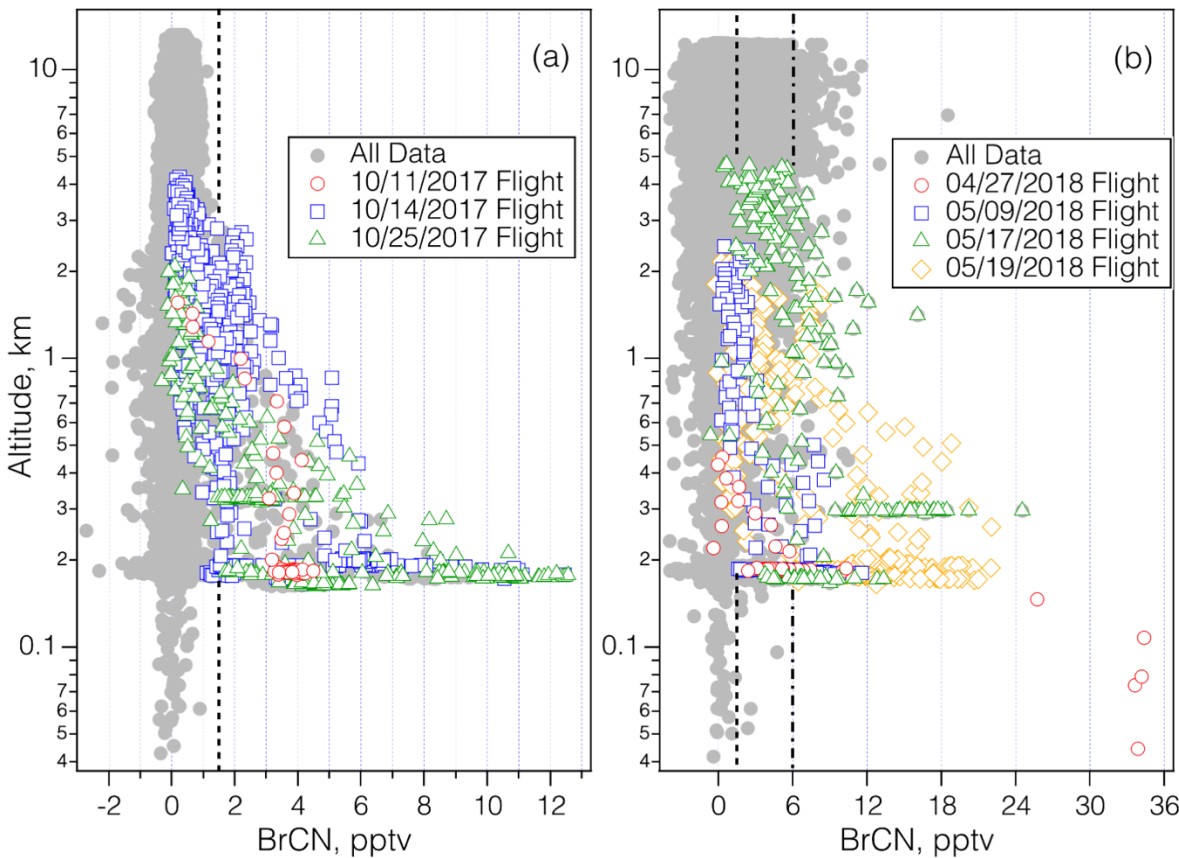

**Figure 3. Vertical profiles of BrCN mixing ratios (10 s averages) measured during ATom-3 (panel a) and ATom-4 (panel b). The colored points correspond to the lowest 4 km of altitude profiles that encompassed polar boundary layer dips on dates noted when enhanced BrCN was observed, and the grey points show the remaining observations. The ATom-3 flights took place in October, 2017 and the ATom-4 flights took place in late April through May, 2018. The 10/11/17 flight took place over the Southern Ocean, the 10/14/17 and 5/09/18 flights took place over the Weddell Sea, and 05/17/18 flight took place over Greenland, and the 10/25/17 and 05/19/18 flights took place over the Arctic Ocean. The doted-dashed line shows limit of detection (LOD) for the flights on May 12, 17 and 21, and the dashed line shows the LOD for the remainder of the flights.**

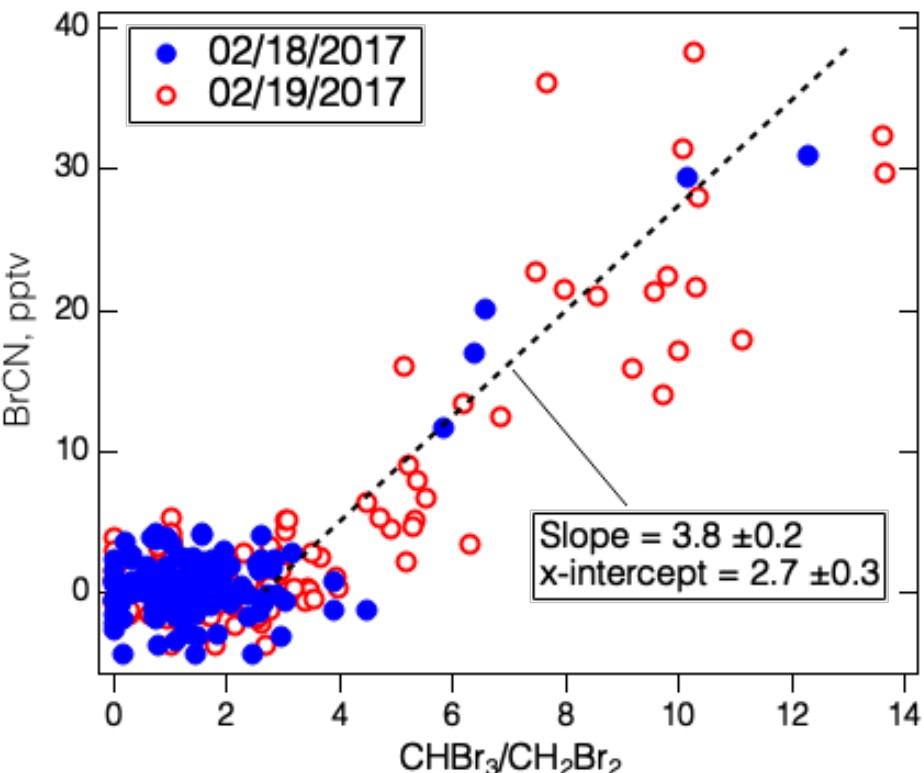

**Figure 4. The BrCN observed by the CIT-CIMS during the ATom-2 Arctic flights versus the ratio CHBr$_3$/CH$_2$Br$_2$ measured by the UCI-WAS. The CIT-CIMS data were averaged over the UCI-WAS sample collection times. The blue points are from the Feb. 18,2017 flight from the Azores to Thule, Greenland, and the red circles are from the Feb. 19, 2017 flight from Thule to Anchorage, Alaska. The dashed line is an iterative fit assuming a background in CHBr$_3$/CH$_2$Br$_2$ and using ODR.**


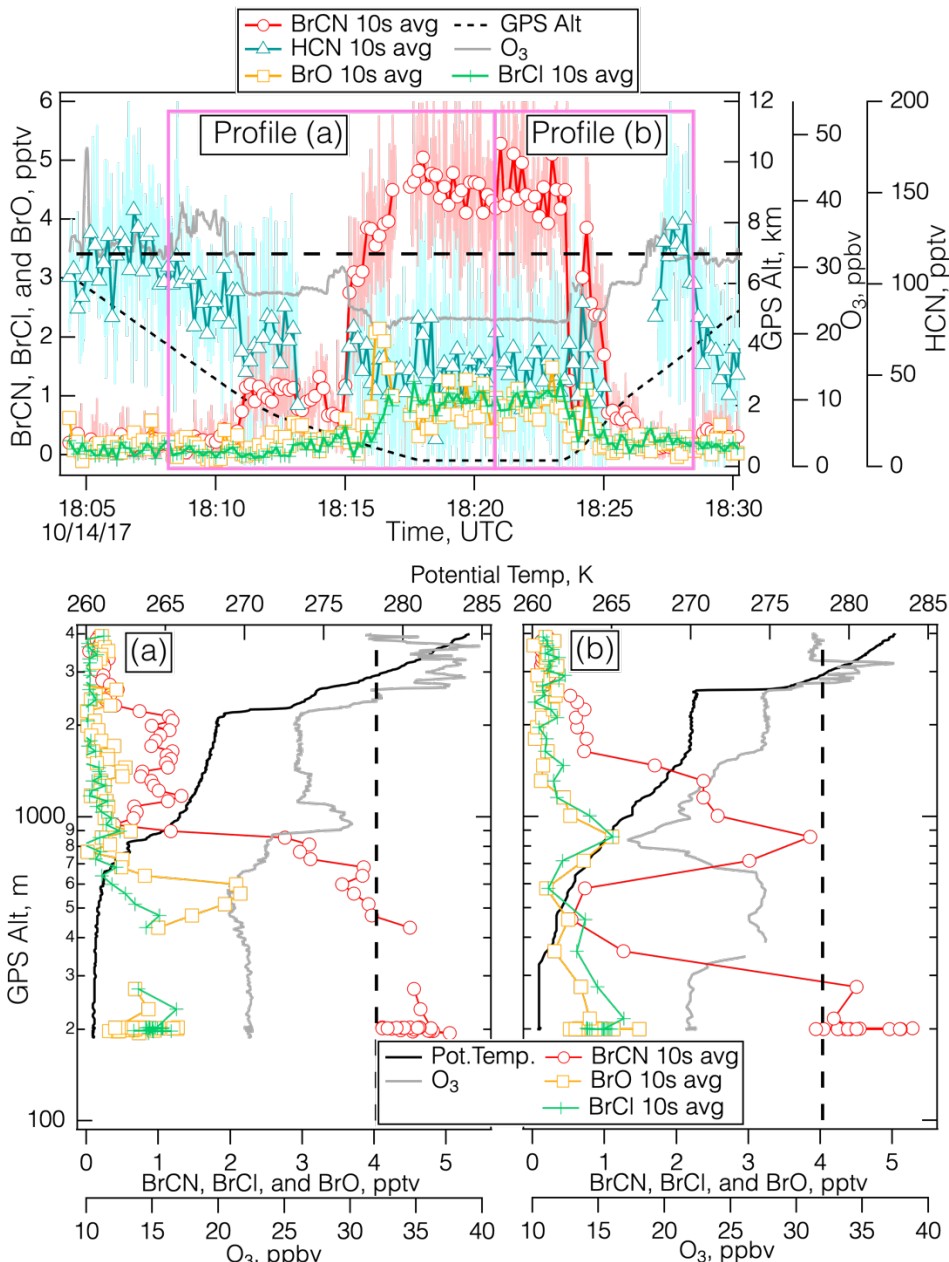

**Figure 5. Details of the measurements during the polar boundary layer leg of the Oct. 14, 2017 flight over the Weddell Sea (75.7° S Lat, 50.1° W Long) centered around 18:20 UTC. The top panel shows the time series for BrCN, BrO, BrCl, HCN, O₃, and altitude, divided into vertical profiles during the descent (Panel a), and the ascent (Panel b), and a dashed line drawn at 32 ppbv O₃.**


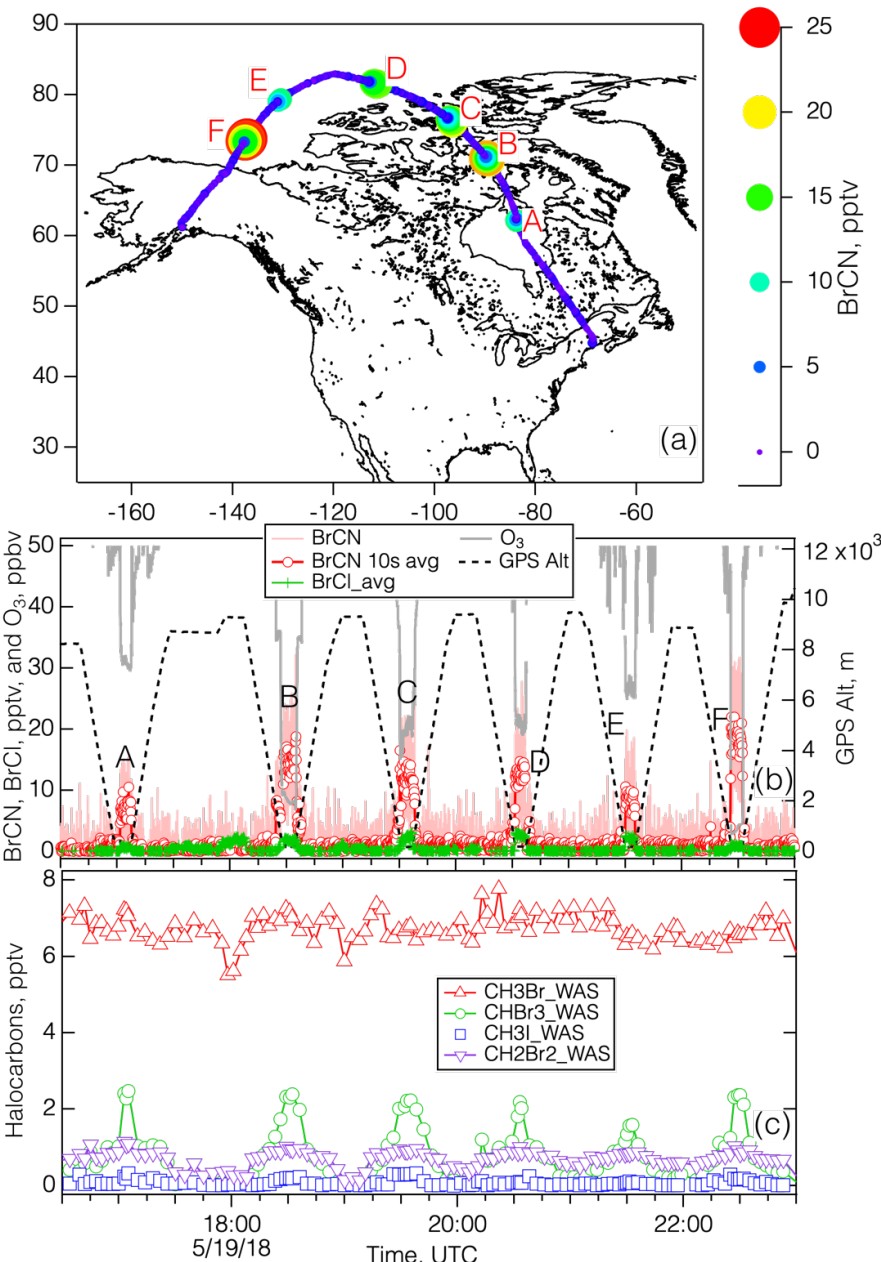

**Figure 6.** Panel (a) shows the map of the May 19, 2018 flight during ATom-4, with the flightpath colored and sized by BrCN, and the polar boundary layer legs labeled A–F. Panel (b) shows the time series of measured BrCN, BrCl and O₃ and the associated altitude and polar boundary layer leg. Panel (c) shows selected halocarbons measured by whole air sampling during the flight. Profile A was measured at 62.2° N Lat, 83.8° W Long, Profile B was measured at 70.9° N Lat, 89.5° W Long, Profile C was measured at 76.3° N Lat, 96.6° W Long, Profile D was measured at 81.4° N Lat, 110.1° W Long, Profile E was measured at 79.2° N Lat, 130.7° W Long, Profile F was measured at 73.4° N Lat, 137.6° W Long.

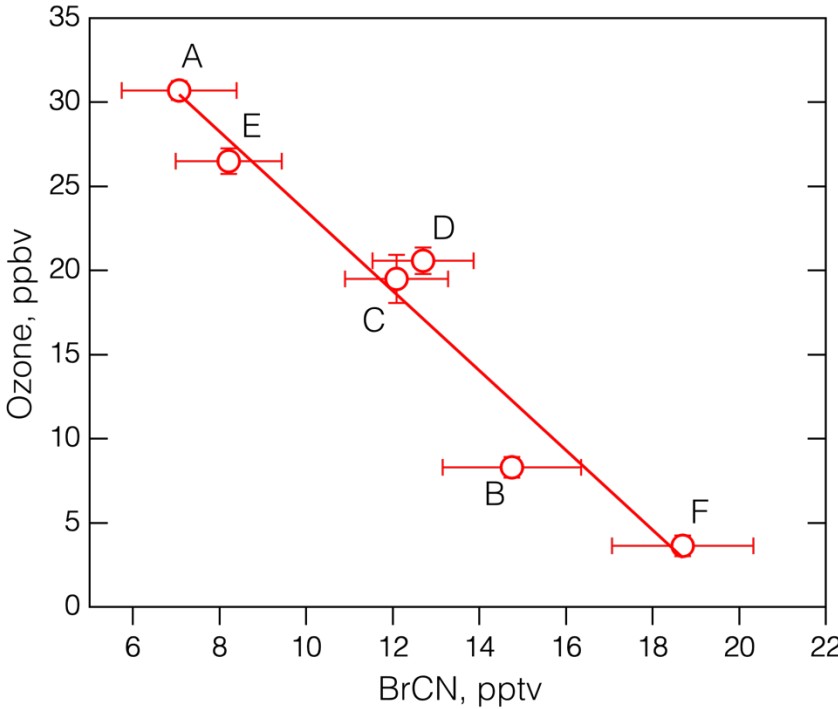


**Figure 7. The average O₃ vs. the average BrCN measured during the polar boundary layer legs of the May 19, 2018 flight of ATom-4. The bars denote the standard deviation of the means and the labels correspond to the periods shown in Figure 5b. The linear regression line drawn through the points has an R² = 0.94.**

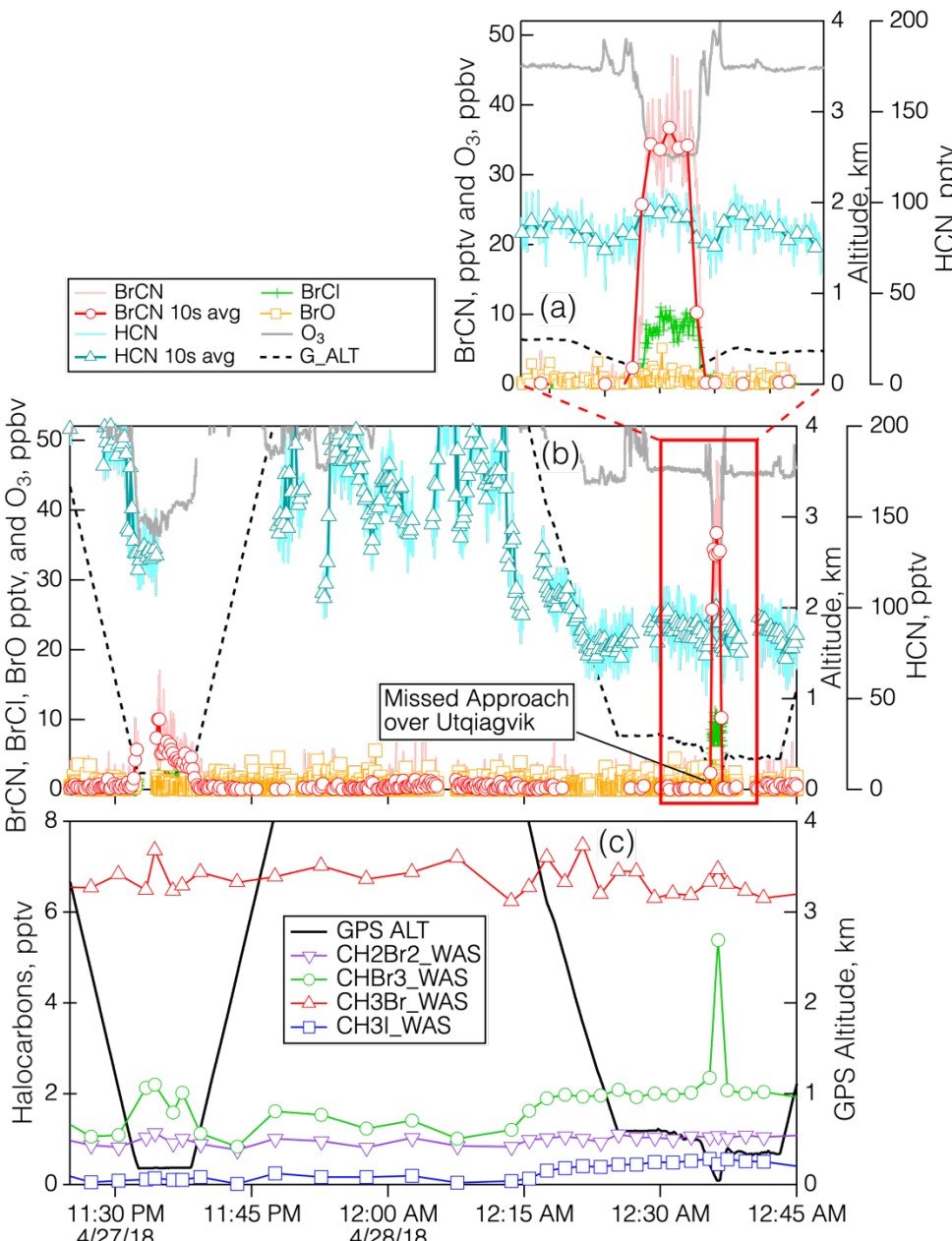

**Figure 8. Details of the measurements during the polar boundary layer legs of the Apr. 27, 2018 flight when the aircraft executed a MA over BRW. Panel (b) shows the time series for BrCN, BrCl, BrO, HCN, O₃, and altitude, and Panel (a) shows the same time series expanded to show the lowest altitudes in more detail. Panel (c) shows the time series of the halocarbons CH₃Br, CH₃I, CH₂Br₂, and CHBr₃. The duration of the WAS samples was shorter than the width of the symbols.**

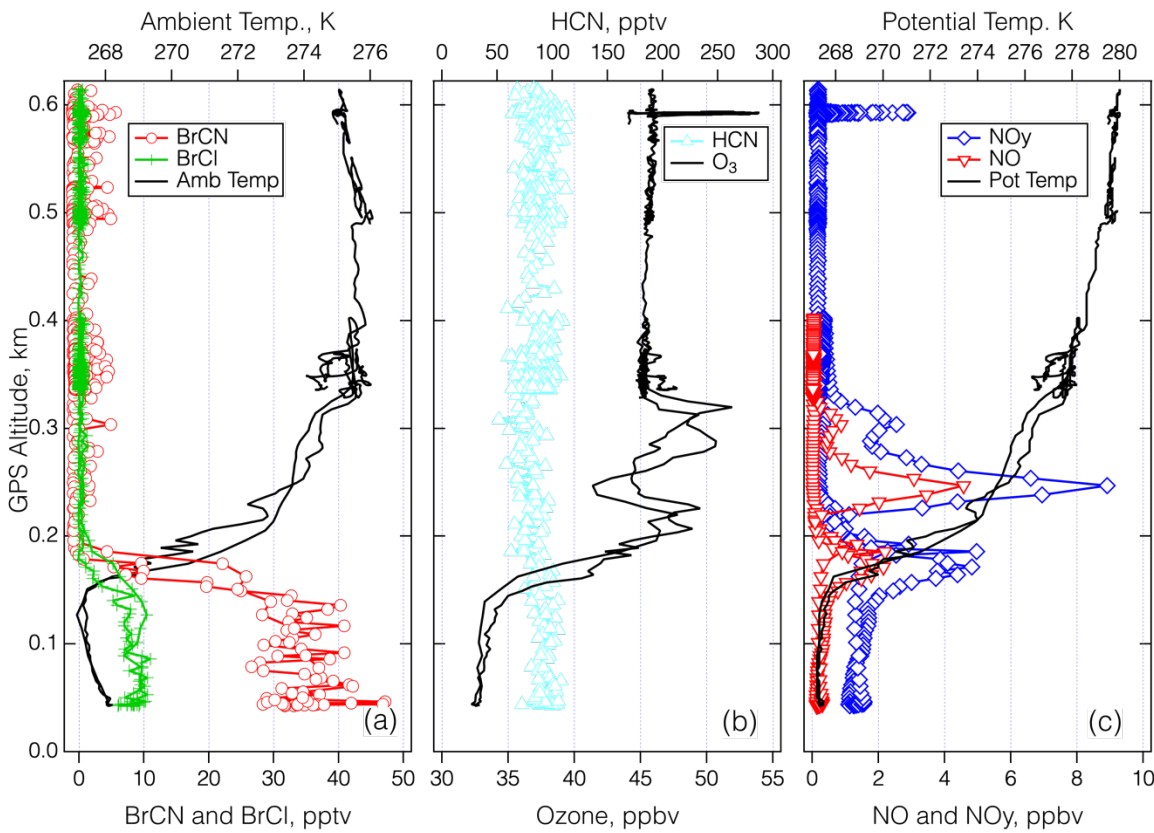


Figure 9. Vertical profiles over BRW on Apr. 27–28, 2018 for the period between 00:25 and 00:43 UTC. Panel (a) shows the 1s measurements of BrCN, BCl, and ambient temperature. Panel (b) shows 1s measurements of HCN and O₃, and Panel (c) shows 1s measurements of NOy, NO and potential temperature.


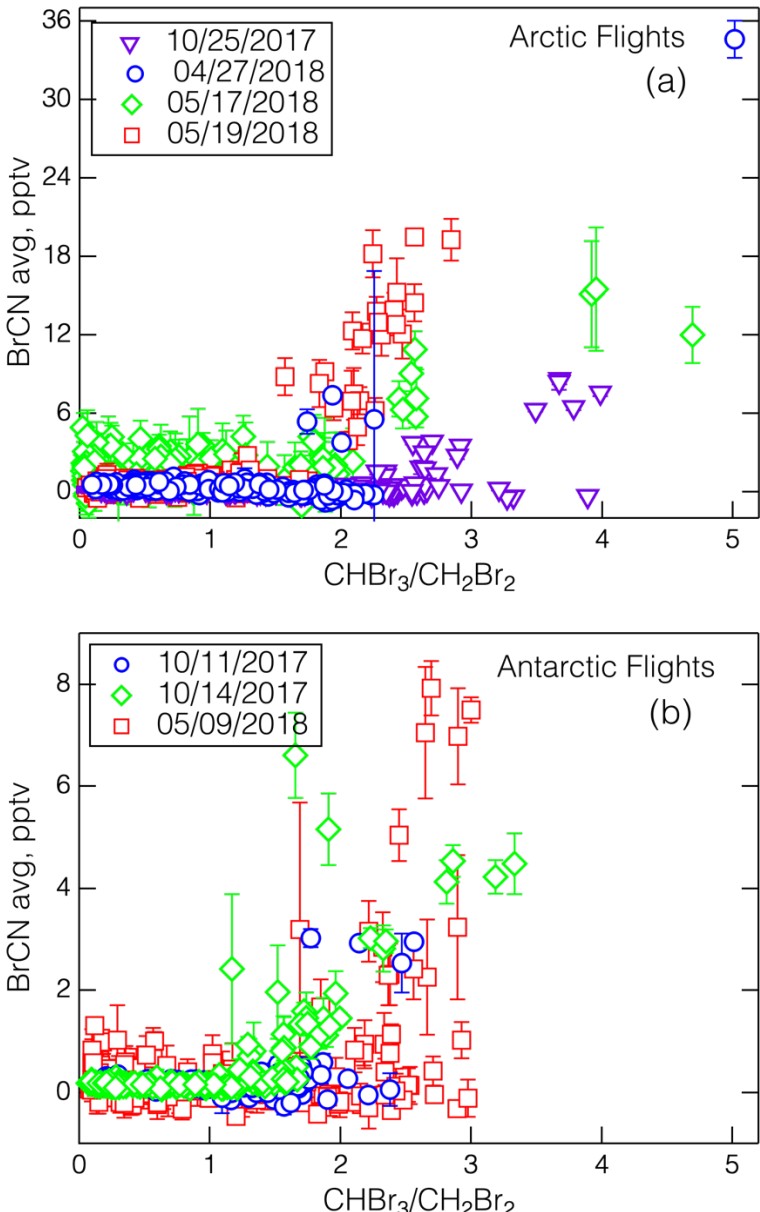

**Figure 10. The relationships between BrCN (10 s data averaged over the start and stop times of the WAS) and the ratio CHBr₃/CH₂Br₂ for the Arctic legs (panel a) and Antarctic legs (panel b), with the corresponding dates as noted, with the standard deviation of the averages shown as error bars. Note the ~~May 09, 2018 and~~ May 17, 2018 data show the effects of degraded sensitivity due to the altered IMR humidity uses on those flights.**


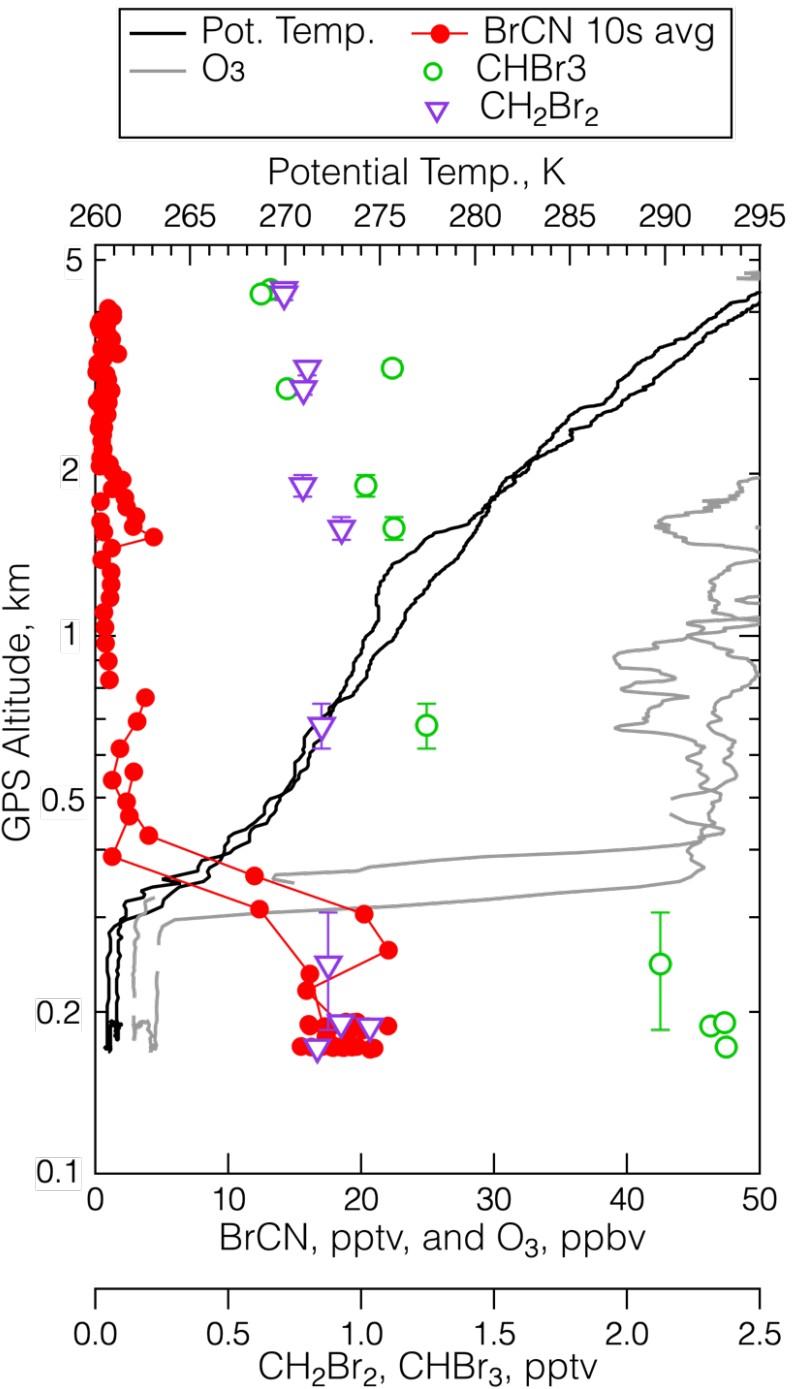

**Figure 11. The vertical distribution of BrCN (red dots), O₃ (grey), CHBr₃ (blue open circles), CH₂Br₂ (purple triangles), and potential temperature (black) during the low-level leg at 22:30 UTC on May 19, 2018 measured at 73.4° N Lat, 137.6° W Long. The error bars are the standard deviation of the average altitude traversed during the WAS sampling time.**