# Peer review of "Observations of cyanogen bromide (BrCN) in the global troposphere and their relation to polar surface O3 destruction."

_EGUsphere, 2023_

## Author Comment (AC1)

Review #1

This is an excellent paper on the chemistry of BrCN, which previously has never been reported to be present in the atmosphere. Detected by two chemical ionization mass spectrometers during the ATom flights, this molecule appears to form in regions where active bromine chemistry is especially prevalent, i.e., in the polar springtime boundary layer when ice/snow is present. The paper presents the measurements, does an excellent job at working through likely BrCN formation and loss pathways, and is well written.

An interesting observation is that high amounts of CHBr3 (relative to CH2Br2) correlate with BrCN. Given that HOBr is believed to form CHBr3 from reactions with DOM, this is indirect support for abiotic formation of BrCN via the multiphase reaction of HOBr with HCN. This makes sense given that this reaction (Reaction S4) has such a large rate constant (close to diffusion limited) in water. HCN is also measured to help constrain the chemistry but an open question, as usual, is the pH of the surface where the multiphase chemistry is occurring. The paper also lays open the potential for there being a biotic source of BrCN (and HOBr).

The main point raised in the paper is that HOBr/Br- chemistry is required for bromine recycling and so, if HOBr is instead reacting to form BrCN, the ozone and mercury loss chemistry shuts down. To my knowledge, this is a new suggestion.

The vertical gradient of the BrCN mixing ratio implies a fairly short lifetime in the atmosphere on the order of days, which is argued to be due to some type of aerosol loss process. Given the high reactivity of BrCN with a range of organic functional groups, it is not unreasonable to hypothesize that complex organobromine compounds are forming as a result.

We thank the Reviewer for the positive comments, and wish to answer the questions and comments as directly as possible.

Questions:

1. Were the calibrations for both CIMS instruments performed with mixing ratios close to the ambient values? That point said, a factor of two agreement between different instruments, ionization schemes, and calibration procedures is pretty darn good for a molecule of this type.

1. The NOAA ToF I-CIMS was calibrated at mixing ratios in the range of 18 - 68 ppbv, and the CIT-CIMS was calibrated for BrCN over the range of 3 - 130ppbv. Both instrument responses

were found to be linear over these ranges. These aspects will be added to the Experimental Methods section.

2. Has anyone ever reported measurements of cyanide ion in ice/snow?

2. We are not aware of any $CN^-$ measurements snow or ice in background polar regions. Conversations with researchers experienced in snow measurements in polar regions (J. Dibb, personal communication, 2023) and a literature search has revealed only reports of contamination of urban snow by ferrocyanide additives used in de-icing salts. This may be because of very low values in general, or due to the fact that $CN^-$ would represent dissolved-N available for biological use (nutrients), hence may not be stable for long periods.

3. Was there any evidence of BrCN in heavily biomass burning impacted regions, where the HCN mixing ratio would be very high?

3. There are two points of comparison that show the BrCN from WF is quite small-to-negligible: BB plumes observed either transported from Siberia (Oct 27, 2017) or off the coast of Africa during the ATom project did not show BrCN above detection limit. Measurements with the same I-CIMS in WF plumes encountered on Aug 3, 2019, during FIREX-AQ (Warneke et al., 2023) showed BrCN to be at most 1 pptv, barely above detection limit (DL $\cong$ 0.3pptv during FIREX-AQ) in 1 min averaged data. These observations were in the middle of plumes in which HCN was over 40 ppbv (measured by the same CIT-CIMS).

4. It is reasonable to assume that HCN is the reactive species for this multiphase chemistry. That said, is it possible that acetonitrile may also be reactive? Probably not, but just wondering.

4. A cursory literature search reveals numerous papers in which reactions of $HOBr/Br_2$ with organic species were carried out in acetonitrile solution. We deduce from this that acetonitrile is essentially inert with respect to reaction with these active Br compounds.

5. My main question: The paper presents data of active bromine species, such as BrCl and BrO. Presumably the CIMS also measured signals for HOBr and Br2. Why were those signals (even if not calibrated) not shown? It would have been interesting to see HOBr/BrCN correlations.

5. Unfortunately, we have found that HOBr and $Br_2$ measurements are not reliable due to interconversion on our inlet (Neuman et al., 2010), and have stated this (lines 184-185, original paper). But we will show the HOBr and $Br_2$ signals for the 5/19/18 flight in our revised manuscript as described in more detail as a response to Reviewer 2.

6. We know from (unpublished) experience that the AMS shows signal for aerosol bromine during ozone depletion events in the Arctic. To my knowledge, these signals have never been calibrated, but it might nevertheless be fun to look at the AMS signals to see if there is any evidence for where some of the Br is going, if indeed it is getting lost via irreversible reactions of organics with BrCN. For example, are there organo-N-Br ion fragments detected in the particles during ozone depletion events?

6. There were no apparent increases in the AMS measurement of non-refractory bromine for the periods when we saw elevated BrCN on Oct 19, 2018, and overall, the measurement was quite low, 1.4 ±0.7 pptv equivalent mixing ratio. We are not aware of any AMS data product that encompassed organo-N-Br ions or related fragments.

My recommendation is to publish this paper after the authors decide whether they want to address the above questions.

We have chosen to address the Reviewer's questions as best we can.

References:

Neuman, J. A., Nowak, J. B., Huey, L. G., Burkholder, J. B., Dibb, J. E., Holloway, J. S., Liao, J., Peischl, J., Roberts, J. M., Ryerson, T. B., Scheuer, E., Stark, H., Stickel, R. E., Tanner, D. J., and Weinheimer, A.: Bromine measurements in ozone depleted air over the Arctic Ocean, Atmos. Chem. Phys., 10, 6503-6514, 10.5194/acp-10-6503-2010, 2010.

Warneke, C., Schwarz, J. P., Dibb, J., Kalashnikova, O., Frost, G., Al-Saad, J., Brown, S. S., Brewer, W. A., Soja, A., Seidel, F. C., Washenfelder, R. A., Wiggins, E. B., Moore, R. H., Anderson, B. E., Jordan, C., Yacovitch, T. I., Herndon, S. C., Liu, S., Kuwayama, T., Jaffe, D., Johnston, N., Selimovic, V., Yokelson, R., Giles, D. M., Holben, B. N., Goloub, P., Popovici, I., Trainer, M., Kumar, A., Pierce, R. B., Fahey, D., Roberts, J., Gargulinski, E. M., Peterson, D. A., Ye, X., Thapa, L. H., Saide, P. E., Fite, C. H., Holmes, C. D., Wang, S., Coggon, M. M., Decker, Z. C. J., Stockwell, C. E., Xu, L., Gkatzelis, G., Aikin, K., Lefer, B., Kaspari, J., Griffin, D., Zeng, L., Weber, R., Hastings, M., Chai, J., Wolfe, G. M., Hanisco, T. F., Liao, J., Campuzano Jost, P., Guo, H., Jimenez, J. L., Crawford, J., and Team, T. F.-A. S.: Fire Influence on Regional to Global Environments and Air Quality (FIREX-AQ), Journal of Geophysical Research: Atmospheres, 128, e2022JD037758, https://doi.org/10.1029/2022JD037758, 2023.

---

## Author Comment (AC2)

Review#2

This manuscript describes the first observation of BrCN in the atmosphere, during the global NASA ATom mission. Measurable BrCN was found within polar (Arctic & Antarctic) boundary layers, and was found in the presence of decreased O3 (relative to background) and increased reactive Br species BrCl and BrO. These measurements are highly novel and provide new understanding of polar bromine chemistry. Detailed comments are provided below and primarily focus on data quantitation, additional experimental information needed, and needed clarifications.

We thank the reviewer for the positive comments about the novelty and importance of this work. We have the following responses to the questions and comments.

In Section 2.3, the authors state that only m/z 190 (and not the second isotope at m/z 192) was used to quantify BrCN (Line 193). They say that they divided the signal by its isotope abundance (0.51) "due to the calibration method employed". Please provide additional details about the calibration described on Lines 195-198. It is confusing then that the calibration was "compared to the sum of...m/z 190 and 192" Page 6, Line 198). It is quite a coincidence that the "slope of the correlation of the CIT-CIMS vs I-CIMS is 0.52 +/- 0.01" (Line 207) (i.e. within error of the isotope ratio), when the CIT-CIMS m/z 190 signal was divided by 0.51. Perhaps there was a simple mistake in the data processing? This calibration is critical to the results presented because this slope led to correction factors being applied to the BrCN data from both the CIT-CIMS and I-CIMS.

We are sorry for the confusion concerning this point. The CIT-CIMS calibration is correct as written. The signals at m/z 190 and 192 were summed during calibration to improve the signal/noise of the analysis, so that the calibration factor included the sum of ion counts from both masses. Subsequent analysis revealed an intermittent interference at m/z 192 (the CIT CIMS had unit mass resolution), so that only m/z 190 could be used for quantitation. As a consequence, that signal at 190 was divided by its fractional isotope abundance before application of the calibration factor that included both ions. This procedure could not be responsible for the original CIT-CIMS data being lower by a factor of 0.52, since the signal that had not been corrected in this manner would have yielded even lower ambient mixing ratios.

Sections 2.2, 2.3, and 2.4.1: Describe the aircraft inlets (lengths, flows, materials, temperatures, etc) used for the two CIMS instruments and chemiluminescence instrument, as many of the compounds being measured can be impacted by line losses. I expect that BrO could have significant line losses, which is why the high flow Eisle-type inlet is typically used on the ground (Liao et al. 2011, JGR, A comparison of Arctic BrO measurements by CIMS and LP-DOAS); this, in particular, is not addressed.

The aircraft inlet for I-CIMS BrCN measurements is partially described by Veres et al., (Veres et al., 2020). Additional information is that the inlet consisted of a 0.75 m length of 0.75cm ID PFA

Teflon tubing thermostated at 35°C and had a flow rate of 6 slpm. The description in the paper will be amended to include this information. Several pieces of evidence argue against significant loss of BrCN inlets: BrCN equilibrates rapidly in the short PFA inlet, consistent with our understanding of the behavior of inorganic species on Teflon inlets as a function of Henry's Law. (see Liu et al., ) (Liu et al., 2019). BrCN hydrolysis is relatively slow and is base catalyzed (Roberts and Liu, 2019), and inlets used during ATom were acidic due to exposure to acidic aerosol (Nault et al., 2020), so we would not expect loss of BrCN due to hydrolysis.

The inlet of the CIT CIMS has been described previously (Crounse et al., 2006;Allen et al., 2022) and consisted of a fluoro-polymer (Fluoropel PFC 801A, Cytonix Corp) coated glass tube under high flow. Consequently, the same aspects of BrCN behavior on surfaces noted above pertain to this inlet as well. We will include this text and associated references in the revised manuscript.

The inlets for the $O_3$ and $NO_y$ chemiluminescence instruments have been described previously (Bourgeois et al., 2020) (Ryerson et al., 1999). The inlet for the NO measurement consisted of a 1.5 m long 4mm ID. PFA tube thermostated at 30°C at a flow rate of 1 slpm, coupled to a fused silica tube 12mm ID, 450mm length, used to match the residence time with the $NO_2$ channel of the instrument. The use of PFA Teflon for NO and $O_3$ sampling is well established and losses of NO or $O_3$ due to chemical reactions would only be expected under extreme circumstances, (sampling of highly polluted air with extremely high VOCs or NO). The $NO_y$ inlet consisted of a solid gold tube situated in the aircraft inlet and heated to 300°C. The loss or incomplete conversion of $NO_y$ species in the $NO_y$ convertor is routinely checked through the addition of $NO_2$ and $HNO_3$ standards, and found to be negligible. We do not use $NO_2$ data in this paper so will not discuss that inlet. We will put these details in the SI.

Losses of BrO on the actual I-ToF CIMS inlet were not apparent during calibrations that were performed after both ATom-3&4 deployments. The intercomparison of in situ BrO measurements by CIMS with Long Path Differential Absorption Spectroscopy (LP-DOAS) reported by Liao et al., (2011) (Liao et al., 2012) provides further evidence that our inlet did not have significant BrO losses. Aside from a large shroud that was required to establish flow from a static air mass, (not required for an aircraft inlet) the key feature of the Liao et al. CIMS inlet was a short section PFA Teflon tubing heated to 40°C and operated at high flow rates. Liao et al., (2011) report excellent agreement between the two methods for periods of steady wind flow without local NO pollution. Our inlet is slightly longer than that CIMS inlet, but was made of the same material and operated at a high flow rate. That study, combined with our laboratory experience indicate that we did not have significant losses of BrO. We will add a short description of this to the experimental methods section of our paper.

BrCN backgrounds were conducted by scrubbing the ambient air (Line 143). Please describe the scrubber (material, temp, etc) and its measured efficiency for removing BrCN and BrCl.

3. The scrubber system consisted of ambient air pumped from separate aircraft inlet, via a diaphragm pump, through a packed trap containing activated charcoal, Purafil, Puracarb AM, and Chlorosorb Ultra, and added to the main inlet periodically as an overflow. The scrubber material

was independently checked for removal of halogens ($Br_2$, $Cl_2$, and BrCl) and other analytes of interest. We have every reason to expect that this scrubber also removed BrCN, however, we have examined zeroing periods that occurred in the middle of BrCN measurements. An example of such a period is shown in Figure R1, from the Oct 19, 2018 flight, in which several of the zero periods happened during elevated BrCN periods. Also, all the zero periods exhibited non-varying signals at or lower than the ambient signals observed. We will add more detailed concerning the zeroing method and will show Figure R1 in the SI of the manuscript.

The authors state concern for sampling inlet reactions, previously reported by Neuman et al. (2010), as the reason why Br2 and HOBr were not quantified. However, the presence of Br2/HOBr, while impacted by sampling inlet reactions, is another clear indicator of active bromine chemistry. Neuman et al. (2010) previously reported Br2 as a lower limit of the sum of HOBr + Br2. I believe that it would be very useful for the authors to report their Br2 and HOBr data, even with high uncertainties, since observations of these species are limited and their presence would provide further support for the current work, especially since the authors propose that HOBr is a precursor to BrCN formation.

4. We will not report HOBr and $Br_2$ data as mixing ratios when we have already explained that we don't have confidence in them. However, we will show a plot of the HOBr and $Br_2$ signals observed during the 5/19/18 flight in the SI of revised paper, as shown in Figure R2 in this response. We note that they are very similar to the BrCl data (which are quantified in our work) in that these active Br compounds are present, but neither correlate closely with the BrCN values, nor anti-correlate closely with ozone during those periods. We think that this is due to the fact that BrO, $Br_2$ and BrCl are all relatively short-lived thus indicate that active Br chemistry is currently happening, while BrCN and $O_3$ loss are more persistent and tend to integrate the amount of active Br chemistry that has happened in an airmass. We will add these points of discussion to the text of the paper in the appropriate section.

The authors discuss the potential for BrCN formation from reactions on the sampling lines. Is there any evidence of the presence of reduced nitrogen compounds on the sampling lines, or is there a simple test that could be done to evaluate this? Were the Br2 and BrO calibrations (noted on Lines 175-176) done using the aircraft inlet or just the instrument, as currently implied? If both HOBr and HCN flow through the inlet, is BrCN formed?

5. We do not have any direct evidence for the presence of reduced nitrogen compounds on the aircraft inlet. There are several lines of evidence that demonstrate that BrCN was not produced from reaction of active Br compounds with reduced nitrogen species on the aircraft inlet. First, a number of BrO and $Br_2$ calibrations were conducted on the aircraft inlet that had been used during ATom-3 but not otherwise cleaned, and no production of BrCN was observed. Second, the NOAA group has conducted preliminary lab tests aimed at examining the temperature and pH dependence of the $Br_2$ + HCN chemistry on ice/water surfaces. These tests were conducted with the quadrupole I-CIMS that had lower sensitivity to BrCN than the ToF CIMS (DL = 30 pptv for

10sec averages). In those tests, combining $Br_2$ (28ppbv) and HCN (5 ppbv) in the PFA Teflon inlet of that instrument (0.4cm ID, 2m length, 2.20 slpm flow of room air at 30% RH) produced no BrCN above detection limit. Substantial production of BrCN was apparent when $Br_2$ and HCN were reacted in a high-surface area reactor containing a small amount of liquid water at pH8, qualitatively confirming the chemistry proposed in this work. We will add a description of this observation in the SI as it pertains to possible instrument inlet effects, however any further description of the BrCN production experiments would be premature.

The methods section (Lines 163-165) states that "clear signals for ICN were observed in ambient air" but were not quantified. Yet, there is no mention of ICN in the Results. It would be useful to learn if ICN was observed concurrently with BrCN, or under different conditions, even it is not quantifiable.

6. We feel the presence of ICN is worth noting, however, we do not want to report observations that we do not have confidence in. Instead, we will add a brief summary to the Results section describing the nature of the ICN observations alluded to in the rest of the paper. We will present the ICN signals during the 5/19/18 flight as a figure in the SI, Figure R3 below, and note that it shows essentially what was described: ICN is sometimes observed when BrCN was observed, but they are not closely correlated. In addition, the instrument background for ICN is relatively high and variable, and the signal in scrubbed air is sometimes slightly higher than in ambient air, often leading to apparent negative ICN mixing ratios. We will show this figure in the SI to support these points. We believe these features, generally high background and apparent negative concentrations, are due to the fact that the reagent ion is iodide, made from the addition of substantial amounts of $CH_3I$, hence the possibilities for ion exchange and surface artefact formation are numerous. We further recommend that alternate ionization schemes (e.g. $Br^-$ ions) would probably yield a better measurement of ICN.

For the ATom-2 Arctic flights on Feb. 18 & 19, 2017, the authors assert on Lines 293-294 that "There was essentially no photosynthetic activity at the Northern latitudes during this time", and state Lines 296-298 that "there was Br activation initiated by O3-Br-chemistry...because there was insufficient photochemistry to carry the gas phase catalytic Br chemistry". This is again repeated in the conclusions on Line 593. Yet, no observational support or reference is provided for these statements. These flights include MAs over BRW and SCC, where Polar sunrise occurs in late January, and by mid-February there are several hours of sunlight. Further, Raso et al. (2017, PNAS) and Custard et al. (2017, ACS Earth & Space Chem.) previously showed photochemical snowpack Br2 and I2 production at the beginning of February to mid-Feb in Utqiagvik (BRW). In addition, Pratt et al. (2013, Nat. Geosc.) showed that ozone reaction with bromide is far less efficient than snowpack photochemistry, and this is supported by the results presented by Custard et al. (2017). The authors should re-evaluate their explanation.

7. We thank the reviewer for making these points. We agree that we need to reevaluate our explanation for the ATom-2 observations. From the actinic flux measurements made on board the

aircraft, we can see that there was sunlight during the Feb 18 &19 2017 flights, although it was much reduced (factors of 10 or more) relative to the ATom-4 periods. Also, temperatures were substantially lower in February relative to May. As a consequence, any photosynthetic sources of BrCN, and $CHBr_3$ (and subsequent biotic conversion to $CH_2Br_2$) were much less important during this ATom-2 period relative to ATom-4. We are also grateful for the reviewer pointing out that the snow pack photochemical source of $Br_2$ is operative during the late Winter period. We believe these features account for the BrCN, and high $CHBr_3/CH_2Br_2$ ratios that were observed during ATom-2. We will change our text in the Results and Conclusions sections to reflect this and will include the suggested references in the process.

In Figure S4, BrCN increases up to ~40 ppt below 600 m and O3 decreases to ~15 ppb in this same altitude range, but this anti-correlation is not clearly described in the text on Line 297. The text also rules out a bromine explosion without O3 below 10 ppbv; yet, BrCl and BrO data from ATom-3 and ATom-4 are discussed without this full ozone depletion, which is inconsistent. Please clarify the text. Also, is information about the boundary layer height available for these data points?

8. It is true that both high BrCN and lower $O_3$ values were observed at the low altitudes in the ATom-2 flights. However, unlike the May 19, 2018 flight there, there was not a close anticorrelation in the ATom-2 BrCN and $O_3$ data when the individual times are considered. We will make this clear in the text when these data are presented. We agree that the two aspects of our text were not in agreement. There is no requirement that $O_3$ be below 10ppbv for the Br explosion chemistry to be happening. We will change the way the ATom-2 data are discussed in the text and simply note that the points from ATom 2 do not closely anti-correlate. We have added the potential temperature data measured on the aircraft, to Figure S4 (see Figure R4 below), and that shows that for most of the low altitude profiles, the potential temperature indicates close to neutral, or stable conditions at about 600m and below.

Abstract: It would be helpful to expand the abstract to briefly explain how the conclusions presented on Lines 26-29 were obtained, including mentioning that box modeling was conducted. Further, it would be helpful to also mention the observed seasonality of the BrCN and its relationship to the other compounds measured that currently missing from the abstract but are key to the study (O3, CHBr3/CH2Br2, BrCl, BrO; also mention in the abstract that these were measured).

9. We apologize, the abstract as submitted was left over from a version of the paper submitted to a journal that had a rather restrictive word limit on abstracts. We agree that it will be helpful to expand the abstract in the manner suggested by the Reviewer. We propose to use this more expansive abstract:

"Bromine activation (the production of Br in an elevated oxidation state) promotes ozone destruction and mercury removal in the global troposphere, and commonly occurs in both springtime polar boundary layers, often accompanied by nearly complete ozone destruction. The chemistry and budget of active bromine compounds (e.g. $Br_2$, BrCl, BrO, HOBr) reflect the cycling of Br and affect its environmental impact. Cyanogen bromide (BrCN) has recently been

measured by iodide ion high resolution time-of-flight mass spectrometry (I⁻ CIMS), and trifluoro methoxide ion time-of-flight mass spectrometry (CF₃O⁻ CIMS) during the NASA Atmospheric Tomography mission 2nd, 3rd and 4th deployments (NASA ATom), and could be a previously unquantified participant in active Br chemistry. BrCN mixing ratios ranged from below detection limit (1.5 pptv) up to as high as 36 pptv (10 s avg) and enhancements were almost exclusively confined to the polar boundary layers in the arctic winter and in both polar regions during spring and fall. The coincidence of BrCN with active Br chemistry (often observable BrO, BrCl and O₃ loss) and high CHBr₃/CH₂Br₂ ratios imply that much of the observed BrCN is from atmospheric Br chemistry rather than a biogenic source. Likely BrCN formation pathways involve the heterogeneous reactions of active Br (Br₂, HOBr) with reduced nitrogen compounds, for example hydrogen cyanide (HCN/CN⁻), on snow, ice, or particle surfaces. Competitive reaction calculations of HOBr reactions with Cl⁻/Br⁻ and HCN/CN⁻ in solution, as well as box model calculations with bromine chemistry, confirm the viability of this formation channel and show a distinct pH dependence, with BrCN formation favored at higher pHs.  Gas phase loss processes of BrCN due to reaction with radical species are likely quite slow and photolysis is known to be relatively slow (BrCN lifetime of ~4 months in mid-latitude summer). These features, and the lack of BrCN enhancements above the polar boundary layer, imply that surface reactions must be the major loss processes. The fate of BrCN determines whether BrCN production fuels or terminates bromine activation. BrCN reactions with other halogens (Br⁻, HOCl, HOBr) may perpetuate the active-Br cycle, however, preliminary laboratory experiments showed that BrCN did not react with aqueous bromide ion (<0.1%) to reform Br₂. Liquid phase reactions of BrCN are more likely to convert Br to bromide (Br⁻) or form a C-Br bonded organic species and so would constitute a loss of atmospheric active Br. Thus, accounting for the chemistry of BrCN will be an important aspect of understanding polar Br cycling."

Both the abstract (Line 29) and conclusions (Lines 615-616) refer to condensed phase loss of BrCN resulting in either Br- or C-Br bonds. What is this based on, and where is it shown in the text?

10. The condensed phase (i.e. solution phase) chemistry of BrCN is relatively well studied, since BrCN is used as a diagnostic reagent in biochemistry, and a reagent in organic synthesis. In fact, the reaction of BrCN with tertiary amines is a name reaction in Organic Chemistry (von Braun and Schwarz, 1902). The requested information was covered in Lines 578-584 in the original paper.

Additional comments:

- Lines 32-37, 42-46, 93-96, 99-101, 188, 329-330, 430-432, 560-565, 608-609: Add references to these sentences.

We will add references where requested, however there are three instances (42-46, 99-101, 560-565, where the reference(s) for these statements are given in the sentence immediately before the lines noted. Perhaps this is a style issue? We will clarify so that it is clear that the statements are supported by the preceding reference(s).

- Line 43: Note that Pratt et al. (2013, Nat. Geosc.) showed that Br2 does not form on sea ice because the surface is buffered (Wren & Donaldson et al 2012, ACP). Also, Wang et al. (2019, PNAS, "Direct detection of atmospheric atomic bromine leading to mercury and ozone depletion") is an appropriate reference to include in this sentence.

We thank the referee for pointing out these features of the $Br_2$ chemistry. They are consistent with our theory that BrCN formation is favored on substrates that have high pH. We will note these aspects of $Br_2$ chemistry at this point and include the suggested references.

- Lines 82-92: It would be helpful if this intro discussion could be clarified, as it connects to later discussion of results. The following papers may be useful to consider incorporating: Swanson et al. 2007 (Atmos. Environ., "Are methyl halides produced on all ice surfaces? Observations from snow-laden field sites"), Rhew et al. 2007 (JGR, "Methyl halide and methane fluxes in the northern Alaskan coastal tundra"), Macdonald et al. 2020 (ACP, "Consumption of CH3Cl, CH3Br, and CH3I and emission of CHCl3, CHBr3, and CH2Br2 from the forefield of a retreating Arctic glacier").

We will clarify the discussion at this point and include the references that the reviewer has pointed out to us. The methyl halides do show some minor production in snow packs that might be associated with photochemistry, or might be biological. The paper on emissions from glacial margins and forefields in particular showed $CHBr_3/CH_2Br_2$ ratios in the range of 1 to 3 (Macdonald et al., 2020). All of these observations are either tangential to, or support our assertion that $CH_2Br_2$ and $CHBr_3$ are the important markers that can distinguish biogenic and abiotic formation.

- Lines 131-132: Please define "MA", "level legs", and other common flight terms here and elsewhere to make the text more readable to the non-aircraft audience.

We will add the following explanations to the text at this point, and continue using the terms 'level leg' and 'MA' in the rest of the manuscript since it has been defined here. The term 'level leg' refers to a flight segment during which the aircraft maintained a constant altitude ($\pm$ a few percent) for at least several minutes or more. The term 'MA' will be defined here as 'missed approach' during which the aircraft descended into the controlled air space above a designated airfield, almost to the point of touching down, and climbed back out.

- Lines 149-150: Please provide information about the PTR-ToF, including the ion(s) used to detect ClCN.

The Hi-Res PTR-ToF instrument has been described in the literature (Yuan et al., 2016;Yuan et al., 2017), and the compound was detected at ClCN•H+, a process greatly aided by the Cl isotope distribution and negative mass defect.

- Lines 153-154: Were the BrCN and ICN in N2?

No, the standards were in zero air, which will be noted here.

- Lines 160-161: Please add these data showing the sensitivity as a function of IMR water vapor concentration to the SI. Also, were the analyte ions normalized to I- or IH2O- prior to calibration?

We will add figure R5, showing the water vapor dependence of the sensitivity to BrCN to the SI. The signals were normalized to $I \bullet H_2O^-$ prior to application of the appropriate sensitivity as will be noted in this section.

- Section 2.3: Describe how the HCN calibration and zero measurements were conducted. How frequent were the "periodic zero measurements"?

The CIT-CIMS instrument was zeroed every 15 minutes in flight by closing instrument to ambient air sampling and adding UHP $N_2$ gas. A second type of zero was also periodically performed by flowing ambient air through a filter containing $NaHCO_3$ coated nylon wool and palladium coated alumina pellets. Both types of zero generally agree well, except at very high ambient RH where HCN is not efficiently scrubbed by the $NaHCO_3$ filter. CIT-CIMS HCN calibration was performed in the lab using a compressed gas mixture of HCN in $N_2$, and a dilution system composed of three mass flow controllers, and additional $N_2$ gas source ($LN_2$ boiloff). FTIR measurements were used to quantify, and monitor the stability of, the HCN mixing ratio in the compressed gas mixture using HITRAN HCN lines strengths. The laboratory HCN calibrations were proxied to flight data using periodic in-flight calibrations (every ~3 hours) of other species ($H_2O_2$, peroxy acetic acid, and ethylene glycol), after accounting for appropriate water vapor and temperature dependencies. We will include this material in the Experimental Methods section.

Lines 197-198: Define "IR" and "MS" acronyms, and check that other acronyms throughout the manuscript are all defined at their first use.

We will define IR (infrared) and MS (mass spectrometry) here, and check that the manuscript to make sure other acronyms are defined at first use.

- Figure 2b: Given the instrument sensitivity issues on several ATom-4 flights (Line 268), perhaps it would be helpful to distinguish data below the limit of quantiation in the figure, as it is more difficult to distinguish the polar enhancement from this figure due to the data points that appear to be below the LOQ.

We have attempted to modify Figure 2b in the manner the Reviewer has suggested by overlaying in grey the points from May 12, 17 and 21, 2018 that were below detection limit. We hope this makes the polar data points more apparent.

- Lines 275 and 278: Please state the dates of these BRW MAs for improved clarity.

We will add the requested information, Oct 2, 2017 for the ATom-3 MA over BRW and Apr 28, 2018, for the ATom-4 MA over BRW.

- Lines 290-301: It would be helpful to add a map of the ATom-2 flights to the SI for context.

We will add a map of the ATom-2 BrCN observations (Figure R7) to the SI.

- Line 333: Please refer to the section where these atmospheric lifetimes are discussed.

We will refer to Section 4.2 here.

- Line 334, Figure 4 slope, Figure S2 slope: Fix significant figures (e.g., should be 26.5 +/- 0.2 ppbv).

We are not sure we understand this point based on the Reviewer's example, as our number in Figure 4 also has 3 significant figures (3.75±0.17). We think the Reviewer's point is that the uncertainty (±0.17) implies that only 2 significant figures are appropriate? We will change Figure 4 to read 3.8±0.2 and Figure S2 to read (0.52 ±0.01) – 0.34.

- Add references for R4-R6, S1-S4, EqS2

R4-R6 are not presented as fundamental reactions but rather to illustrate that active bromine reactions are in competition, the rate constants where available can be found in the literature described in the introduction. We note that we had mis-labeled the reaction that appeared on Line 572, it should have been R7, but it also has no reference as such, since it is being hypothesized here. We think this is clear from the text following R7. The reference for S1-S4 (Gerritsen et al., 1993) will be given in that section. We will add a good general reference for S2, effective Henry's Law constants (Sander, 2015).

- Line 328-329: Discuss altitudes and boundary layer heights here for context since the figure is in the SI.

We will now note that the layer in which the BrCN was observed extended up to 0.8km.

- Lines 335-336: Define the "intermediate polar boundary layer" and "middle of the polar boundary layer".

We would prefer not to try to discuss this here, as it is presented in the Discussion section, (see the paragraph starting on line 508 of the original manuscript). Instead, we will add a sentence "See the Discussion section for a further description of the polar boundary layer."

- Line 349: Clarify whether the Southern Ocean was frozen.

Unfortunately, video from the DC-8 is not available for that flight. The TERRA/MODIS imagery from that date (NASA and Worldview, 2018) and nearby dates indicates that the Weddell Sea was largely frozen, but we cannot be sure there were no open leads. We will add a sentence to this section about this.

- Lines 356-360: Refer to where these data are shown. If not currently included, please add as a figure in the SI.

We will add figures to the SI (figures R8 and R9 below).

- Lines 380-381: The authors are encouraged to also consider the work of Halfacre et al. (2014, ACP, "Temporal and spatial characteristics of ODEs from measurements in the Arctic") here.

We thank the reviewer for suggesting the paper, (Halfacre et al., 2014) which provides further evidence of the widespread nature of ODEs, and we will also add the reference for Ridley et al., who showed widespread ODEs in the Arctic springtime in the year 2000 (Ridley et al., 2003). We will include these in our revised paper at this point.

- Lines 498-499: Note that this statement does not appear to consider aldehydes or ketones.

No, we do not explicitly mention aldehydes and ketones. The point of the text in these lines was to note the condensed phase reactions for which we have rate constants and to examine the implied atmospheric lifetimes in relation to these rates. A search of the literature turns up a number of papers in which BrCN reacts with β-diketones by substitution of Br at the central carbon (ie. forming a C-Br bond) and a paper that describes a general route to epoxides (Li and Gevorgyan, 2011). The key point of this section is that we have not found any reactions in which BrCN reforms active-Br. We feel that any more extensive summary of BrCN condensed phase chemistry is beyond the scope of this work.

- Section 4.1: The modeling of the pH dependence of BrCN production (Figures S17 and S18) is noteworthy, and these results warrant more discussion in the main text.

We will present more discussion of this feature and in the process discuss some of the pH-dependent processes noted by this Reviewer above.

- Line 513: It is stated that "there are often clouds in the vicinity of the inversion", but no reference is provided for this statement.

This is discussed is in one of our references (Serreze et al., 1992) and will be noted here.

- Lines 524, S61-62: Fix reference formatting.

These will be fixed.

- Lines 566-568: Describe the details of this laboratory experiment in the SI.

We will put a brief description of this experiment in the SI. The reactor and flow system was similar to that described previously (Roberts et al., 2009) and the instrument responses were calibrated via $Br_2$ permeation tubes and the BrCN diffusion source.

- Lines 596-597: Provide the date for context.

These will now be added to the text at this point.

- Lines S55-S56: Are both [Cl-] and [Br-] assumed to be 16 uM? No reference is provided, and previous measurements by Krnavek et al (2012, Atmos. Environ.) suggests that this assumption is not appropriate.

The 16uM refers to the sum of Cl- and Br-. This is at the low end of the distributions of these ion concentrations measured in snow and ice. We will note that and include the suggested reference (Krnavek et al., 2012), and further references to support this point. Higher halide concentrations would have the effect of shifting the blue dashed line in Figure S17 upwards, and we will note that in the SI text.

- Line 598: Change "ice and particle" to "ice, and/or particle".y

This will be changed.

- Line 605: Note that the sea ice surface is buffered (Wren & Donaldson et al 2012, ACP).

We will include this in the text at this point.

- Figure 1: State in the caption that this was during ATom-3.

We will include this in the Figure caption.

- Figure 3: In the caption, provide the full date ranges for ATom-3 and ATom-4 for context, and also provide the general locations for the highlighted flights. Since a significant fraction of the data are below LOD and LOQ, please state these in the caption for context, or perhaps draw lines in the figure.

We will include the full range of dates of ATom3&4 in the figure caption, and provide the general locations for the dates that are shown explicitly. We will modify the figures to include lines for the LODs.

- The data in many of the figures is below LOD and/or LOQ, it would be helpful to indicate these values in the captions or show more clearly in the figures.

We will repeat the LODs and dates in the figure captions for Figures 2&3, which are really the only ones where all the data are summarized.

- Figures 11, S7, S9, S11, S19: In the captions, state the location where these data were collected.

Much of this requested information is redundant, for example the 22:30 UTC leg on May 19 is shown in several figures and the profiles shown in Figure S11 are at the locations shown in Figure 6a. Nevertheless, we will put the latitudes and longitudes in the figure captions. We also added this information to the caption for Figures 5&6.

- Figure S2: Add error to the slope fit through 0.

The uncertainty (±0.008), will be noted in the figure.

- Table S1: Define abbreviations in the caption.

The abbreviations in the Table title will be spelled out.

Table S2: My understanding from Line 298 is that the LOD for BrCN during ATom-4 was higher. Please clarify that here. Also indicate the m/z used for quantitation of each compound.

As we stated in the experimental approach section, there were 3 flights for which the instrument sensitivity was degraded due to the use of higher water vapor in the IMR. We will note that again here.

Figures

[Figure]

Figure R1. The time series of I$^{79}$BrCN$^-$ signals measured during the 5/19/18 flight over the Arctic. The grey lines are measurements of ambient air and the black lines are measurements during times the inlet was over-flowed with scrubbed air.

[Figure]

Figure R2. The time series of IBr$_2^-$ and IHOBr$^-$ signals (in normalized counts per second) along with GPS altitude and O$_3$ for the flight on May 19, 2018.

[Figure]

Figure R3. Comparison of the I•ICN⁻ signals observed on 5/19/18 with those of I•BrCN⁻. The grey lines are signals from ambient air and the black lines are signals during the periods when the inlet was overflowed with scrubbed air.

[Figure]

Figure R4. The ATom-2 BrCN and O₃ measurements as a function of altitude for both Arctic flights. The measurements were averaged over the UCI-WAS sample time to be compatible with

the data in Fig. 4 in the main text. The BrCN values are shown as red circles and the $O_3$ values are shown as black crosses, and the potential temperature is shown as a black line.

[Figure]

Figure R5. The water vapor dependence of I-CIMS sensitivity to BrCN.

[Figure]

Figure R6. Modified Figure 2b with the observations on May 12, 17 and 21, 2018 that were below detection limit shown in grey.

[Figure]

Figure R7. The map of BrCN mixing ratios (60s avg) observed during the Feb 18 and Feb 19, 2017 flights during ATom-2.

[Figure]

Figure R8. The time series of bromocarbons (Panel a) and BrCN, BrCl, HCN, $O_3$ and altitude (Panel b) for the first period in May, 09, 2018 when BrCN was encountered.

[Figure]

Figure R9. The time series of bromocarbons (Panel a) and BrCN, BrCl, HCN, $O_3$ and altitude (Panel b) for the second period in May, 09, 2018 when BrCN was encountered.

References:

[revised manuscript text omitted]

---

## Author Response (AR2)

The authors' revisions have significantly improved this manuscript focused on the first observations of atmospheric BrCN. The majority of the reviewer comments have been addressed. In particular, the expanded abstract and many figures added to the SI are significant improvements. In addition, critical sampling and measurements details have been added to the Methods section. The remaining comments mainly focus on clarifications and revisions needed to the added/revised text. I refer below to line numbers in the ATC version of the manuscript.

We thank the reviewer for their thoughtful comments and we have the following responses (noted in different font), and have made changes to the manuscript (noted in red).

L227-236 (new text): The authors added the statement: "Losses of BrO on the actual I-ToF CIMS inlet were not apparent during calibrations that were performed after both ATom-3 and -4 deployments." More information is needed here about the calibrations, including reporting the calibration factors and the actual loss %s observed. The assertion that their inlet did not experience losses based on Liao et al. (2011), who used a different inlet, is not acceptable. They base this on both being operated "at a high flow rate" (L233 & 235). On comparison, the Liao et al. inlet had a flow rate of 900 slpm through a 7.6 cm ID aluminum pipe, followed by 13 lpm through a 25 cm, 0.65 cm ID Teflon tube. Therefore, this inlet is very different from the authors' aircraft inlet that was 0.75 m, 0.75cm ID PFA Teflon tubing at 6 slpm (L169-170). From personal experience using the Liao et al. and other inlets, we have observed major BrO losses (far greater than Br2/HOBr problems) when the high flow through the large pipe is not used – and when only smaller diameter perfluoropolymer tubing is used. Further, the authors argument is inconsistent in that the inlet described by Liao et al. has been used for quantitative Br2 and HOBr measurements (including by coauthor – Wang et al. (2019) PNAS), whereas the authors assert that their inlet is not reliable for Br2 and HOBr (L236-237), showing that their inlet experiences increased losses compared to the design described by Liao et al. Therefore, I urge the authors to: 1) remove the added text on L229-234 that justify minor losses on their inlet based on Liao et al., 2) add %s BrO loss measured (and information about that test), and 3) add a statement that losses of BrO may have occurred on the inlet.

There are two factors at work in the inlet described by Liao et al (2011): initiation of flow from a stagnant airmass into a fast flow inlet, and behavior of BrO once it is in a Teflon inlet that is sampling off of that fast flow inlet.

As described by Liao et al., 2011 initiation of flow next to a structure requires a shroud and a very large flow in order to avoid sampling air that has had significant contact with surfaces of the structure. Even with this configuration, it's really the air that is in the core of the flow that is unperturbed by surface losses. It is therefore unsurprising that the Reviewer noted significant

BrO losses when the outer inlet was not used. The aircraft inlet used in our work and in the work of Liao et al., 2012a have no such flow initiation problems, as the aircraft was traveling at 100 m/s, or faster, and the Teflon inlet extended into that flow.

This leaves us with issue of possible losses of BrO on the Teflon inlets. The materials of the two inlets are essentially the same, but the residence times are somewhat different, which led us to use the experience of Liao et al. 2011 as evidence for our inlet. However, we have looked into the details of our laboratory tests on BrO loss and can now present quantitative information to demonstrate that our inlet does not have significant BrO loss.

The issues with $Br_2$ and HOBr that have been noted with our inlet are due to interconversion of these species especially in the presence of surface $Br^-$ ions. This is an effect quite independent of BrO chemistry, as that is either radical-radical chemistry or BrO aqueous-phase chemistry(Liao et al., 2012a), and the tests that we have done in between and after ATom deployments indicated no significant losses of BrO. This independence of BrO from $Br_2$ and HOBr chemistry is in agreement with previously reported tests on a PFA inlet that was slightly longer and of lower flow than our inlet used on ATom (Liao et al., 2012a).

As a result of this discussion about flow initiation, the similarity of the Teflon inlets between our study and (Liao et al., 2012a; Liao et al., 2011), and the results of our BrO tests: 1) we feel some modification, not elimination, of the text in Lines 229-234 is warranted, 2) we can now add that at least 90 ±4% of BrO is transmitted through the ATom inlet, 3) we prefer to state "we have not observed evidence of any other losses of BrO on the ATom inlet".

The added/modified sentences from Lines 228-236 are:

Significant losses of BrO on the actual I-ToF CIMS inlet were not apparent during calibrations that were performed before and after both ATom-3 and -4 deployments. In addition, the intercomparison of *in situ* BrO measurements by CIMS with Long Path Differential Absorption Spectroscopy (LP-DOAS) reported by (Liao et al., 2011), and the tests reported by Liao et al., 2012a provide further evidence that once flow into a Teflon inlet is properly initiated, an inlet of our type would not have significant BrO losses. Aside from a large shroud that was required to establish flow from a static air mass (Liao et al., 2012b), (not required for an aircraft inlet) the key feature of the Liao et al. (2011) CIMS inlet was a short section PFA Teflon tubing heated to 40°C and operated at high flow rates. Liao et al., (2011) report excellent agreement between the two methods for periods of steady wind flow without local NO pollution. Our inlet is slightly longer than that CIMS inlet, but was made of the same material and operated at a high flow rate.

Laboratory tests were conducted during ATom calibrations to assess the amount of loss of BrO in the instrument inlet used in ATom. Those tests compared the signals observed when adding the BrO source directly to the IMR, with those observed with the addition of the 0.7m long ATom inlet operated at 6.3 SLPM. The results showed the ATom inlet transmitted 90 ±4% of BrO, leading us to conclude there was minimal loss of BrO. We further note that this inlet loss was accounted for in the applied calibration factor, determined with the ATom inlet in place. We have not observed evidence of any other losses of BrO on the ATom inlet.

L226-228: The authors cite Neuman et al. (2010) and Osthoff et al. (2008) for the calibration of the I-ToF CIMS, but those prior papers report

calibration of the quad-CIMS. While the approach can be the same, the calibration results are expected to be different between the two CIMS instruments and also are expected to vary based on H2O addition and potentially RH, which needs to be addressed.

The reviewer is correct that these previous papers involved Quad -CIMS instruments with different IMR configurations, and we see by the way this sentence is worded that the reviewer could be mis-led into thinking that calibration factors from these references were used somehow in the current work, when really it is only the calibrations methods (i.e. sample preparation and independent determinations) that were used to calibrate the ToF-CIMS. Therefore, none of the requested information on the quad instruments is relevant. Instead, we will rewrite this sentence to make it clear we are referring to the methods of how the ToF-CIMS was calibrated and include further references about that. The sentence will now read:

Methods for the calibration of the I-CIMS for those species have been described previously(Neuman et al., 2010; Osthoff et al., 2008; Wild et al., 2014) and were applied to the calibration of the ToF-CIMS used in this work.

L175-180 (new text): Please report the measured scrubber efficiency (%) for the tested compounds – Br2, Cl2, and BrCl. Was the scrubbing efficiency measured for BrCN in the lab? I note that added Figure S2 is useful.

We have now added to the sentence on line 176:
and found to be more than 99% efficient at removing these dihalogens.

We did not check BrCN removal in the laboratory, but instead presented Figure S2, which essentially shows that the BrCN was removed at better than 92% efficiency. We have a sentence at lines 179 to read:
The signals during those zeroing periods indicate that the scrubber was at least 92% efficient at removing BrCN.

L379 (new text): Raso et al. (2017, PNAS) showed Br2 and I2 snowpack production, but I2 was not measured by Custard et al. (who also measured snowpack BrCl production, which may be relevant here) or Pratt et al., cited here. Please add Raso et al. for the mention of I2.

We have added Raso et al.., 2017 to this list of references.

Minor Comments:
Line 101 (new text): Fix year (reference listed as published in 2077!).

The date on this reference should be 2007 and has now been fixed.

Line 287: NOy includes NOx, so this header should be corrected to NOy

instead of NOxy.

We are sorry for the confusion. The term 'NOxy' is our short-hand for this instrument, so its use here is inappropriate. The heading for the section is now changed to:

$NO_y$, $NO_x$ and $O_3$ measurements.

**Line 572 (new text): Fix typo and clarify sentence.**

We have fixed the typos and added some clarifying text. The sentence now reads:

Higher $Cl^-$ and $Br^-$ concentrations will serve to shift the pH at which BrCN becomes competitive to higher values, but the range still appears to be 6.5-8.5 pH units, a range that is applicable to matrices in polar environments.

**Line 699 (new text): Cite Custard et al. (2017, ACS Earth & Space Chem) here for this statement, since it is not a result of this study.**

We have now cited Custard et al., 2017 at this point.

References

Custard, K.D, Raso, A. R. W., Shepson, P. B., Staebler, R. M., and Pratt, K. A.: Production and release of molecular bromine and chlorine from the Arctic coastal snowpack, ACS Earth Space Chem., 1, 142-151, 10.1021/acsearthspacechem.7b00014, 2017.

Liao, J., Huey, L. G., Scheuer, E., Dibb, J. E., Stickel, R. E., Tanner, D. J., Neuman, J. A., Nowak, J. B., Choi, S., Wang, Y., Salawitch, R. J., Canty, T., Chance, K., Kurosu, T., Suleiman, R., Weinheimer, A. J., Shetter, R. E., Fried, A., Brune, W., Anderson, B., Zhang, X., Chen, G., Crawford, J., Hecobian, A., and Ingall, E. D.: Characterization of soluble bromide measurements and a case study of BrO observations during ARCTAS, Atmos. Chem. Phys., 12, 1327-1338, 2012a.

Liao, J., Huey, L. G., Tanner, D. J., Flocke, F. M., Orlando, J. J., Neuman, J. A., Nowak, J. B., Weinheimaer, A. J., HAll, S. R., Smith, J. N., Fried, A., Staebler, R. M., Wang, Y., Koo, J.-H., Cantrell, C. A., Weibring, P., Walega, J., Knapp, D. J., Shepson, P. B., and Stephens, C. R.: Observations of inorganic bromine (HOBr, BrO, and $Br_2$) speciation at Barrow, Alaska, in spring 2009, J. Geophys. Res. Atmos., 117, D00R16, 2012b.

Liao, J., Sihler, H., Huey, L. G., Neuman, J. A., Tanner, D. J., Freiss, U., Platt, U., Flocke, F. M., Orlando, J. J., Shepson, P. B., Beine, H. J., Weinheimer, A. J., Sjostedt, S. J., Nowak, J. B., Knapp, D. J., Staebler, R. M., Zheng, W., Sander, R., Hall, S. R., and Ullmann, K.: A comparison of Arctic BrO measurements by chemical ionization mass spectrometry and long path-differential optical absorption spectroscopy, J. Geophys. Res. Atmos., 116, 2011.

Neuman, J. A., Nowak, J. B., Huey, L. G., Burkholder, J. B., Dibb, J. E., Holloway, J. S., Liao, J., Peischl, J., Roberts, J. M., Ryerson, T. B., Scheuer, E., Stark, H., Stickel, R. E., Tanner, D. J., and Weinheimer, A.: Bromine measurements in ozone depleted air over the Arctic Ocean, Atmos. Chem. Phys., 10, 6503-6514, 2010.

Osthoff, H. D., Roberts, J. M., Ravishankara, A. R., Williams, E., Lerner, B. M., Sommariva, R., Bates, T. S., Coffman, D., Quinn, P. K., Dibb, J. E., Stark, H., Burkholder, J. B., Talukdar, R. K., Meagher, J., Fehsenfeld, F. C., and Brown, S. S.: High levels of nitryl chloride in the polluted subtropical marine boundary layer, Supplementary Information, Nature Geoscience, 1, 324-328, 2008.

Raso, A. R. W., Custard, K. D., May, N. W., Tanner, D., Newburn, M. K., Walker, L., Moore, R. J., Huey, L. G., Alexander, L., Shepson, P. B., and Pratt, K. A.: Active molecular iodine photochemistry in the Arctic, Proc Natl Acad Sci., 114, 10053-10058, 2017.

Swanson, A. L., Blake, N. J., Blake, D. R., Rowland, F. S., Dibb, J. E., Lefer, B. L., and Atlas, E.: Are methyl halides produced on all ice surfaces? Observations from snow-laden field sites. , Atmos. Environ., 41, 5162-5177, 10.1016/j.atmosenv.2006.11.064, 2007.

Wild, R. J., Edwards, P. M., Dube, W. P., Baumann, K., Edgerton, E. S., Quinn, P. K., Roberts, J. M., Rollins, A. W., Veres, P. R., Warneke, C., Williams, E. J., Yuan, B., and Brown, S. S.: A measurement of total reactive nitrogen, NOy, together with $NO_2$, NO, and $O_3$ via cavity ring-down spectroscopy, Environ. Sci. Technol., 48, 9609-9615, 2014.